# Acute stress promotes effort mobilization for safety-related goals
Kristína Pavlíčková [1], Judith Gärtner[1], Stella D. Voulgaropoulou[1], Deniz Fraemke[2], Eli Adams[1], Conny W.E.M. Quaedflieg[3], Wolfgang Viechtbauer[1] & Dennis Hernaus [1] ✉

Although the acute stress response is a highly adaptive survival mechanism, much remains unknown about how its activation impacts our decisions and actions. Based on its resource-mobilizing function, here we hypothesize that this intricate psychophysiological process may increase the willingness (motivation) to engage in effortful, energy-consuming, actions. Across two experiments ($n = 80$, $n = 84$), participants exposed to a validated stress-induction protocol, compared to a no-stress control condition, exhibited an increased willingness to exert physical effort (grip force) in the service of avoiding the possibility of experiencing aversive electrical stimulation (threat-of-shock), but not for the acquisition of rewards (money). Use of computational cognitive models linked this observation to subjective value computations that prioritize safety over the minimization of effort expenditure; especially when facing unlikely threats that can only be neutralized via high levels of grip force. Taken together, these results suggest that activation of the acute stress response can selectively alter the willingness to exert effort for safety-related goals. These findings are relevant for understanding how, under stress, we become motivated to engage in effortful actions aimed at avoiding aversive outcomes.

*"These changes-the more rapid pulse, the deeper breathing, the increase of sugar in the blood, the secretion from the adrenal glands-were very diverse and seemed unrelated. Then, [...] the idea flashed through my mind that they could be nicely integrated if conceived as bodily preparations for supreme effort in flight or fighting."*

-*Walter Bradford Cannon (1945)*[1]

The acute stress, also known as fight-or-flight, response is essential for surviving encounters with threats. The release of neurotransmitters and glucocorticoids from major stress axes temporarily suspend homeostatic setpoints in lieu of short-term survival[2,3]. At its core, and as epitomized by the above quote, this highly complex psychophysiological response liberates the necessary resources—energy—that allow us to successfully face or avoid threats[4].

Critically, surviving encounters with threats does not only require the availability of energy; it equally depends on psychological skills that allow us to optimally direct that energy towards survival[3,5,6]. To date, little remains known about how the activation of major stress axes impacts the cognitive

mechanisms that facilitate decisions to expend energy in the service of maximizing safety (e.g., avoiding pain, injury, loss, or negative emotional states). Here, we address this question by casting the acute stress response into the framework of effort-based decision-making and subjective value computations. Specifically, we investigated how experimental induction of acute stress impacted participants' decisions about which threats they deem "worth" avoiding by exerting physical effort.

Stress-induced changes in core psychological domains, including attention, (working) memory, learning[7–10], and underlying neural mechanisms (e.g., large-scale brain network reconfigurations[11]), have been extensively investigated. Few studies to date, however, focused on alterations in the willingness to exert effort—that is, the mobilization of physical or cognitive resources—in pursuit of a goal[12–14]. A core tenet of effort-based decision-making is that exerting effort is inherently aversive and, therefore, preferably avoided[15–17]. Effort reduces the value of desirable outcomes (i.e., effort discounting) and, when faced with cost-benefit dilemmas, humans tend to compute a subjective decision value (i.e., reward—effort cost) to decide if they should exert effort[18–20]. Although primarily investigated in the reward domain, such principles should also

[1]Department of Psychiatry and Neuropsychology, Mental Health and Neuroscience (MHeNs) Research Institute, Maastricht University, Minderbroedersberg 4-6, 6211 LK Maastricht, The Netherlands. [2]Max Planck Research Group Biosocial—Biology, Social Disparities, and Development, Max Planck Institute for Human Development, Berlin, Germany. [3]Department of Neuropsychology and Psychopharmacology, Faculty of Psychology and Neuroscience, Maastricht University, Maastricht, The Netherlands. ✉e-mail: dennis.hernaus@maastrichtuniversity.nl

apply to deliberative decisions involving threats[21]: one should weigh the (e.g., energetic) costs and benefits (e.g., prevented harm) to decide which threats are worth avoiding. Based on the above work, it stands to reason that acute stress may promote survival by altering our willingness to exert effort in the service of avoiding encounters with threats.

Investigating the impact of acute stress on the willingness to exert effort has the potential to advance our understanding of the motivational processes that shape defensive behavior. By eliciting competition between (avoidance of) effort and threats (see below), we aimed to reveal how we become motivated to carry out actions that principally serve to position oneself away from threats. Such behavior can be interpreted as a sign of aversive motivation; a willingness to avoid aversive outcomes, which is thought to be mediated by the fight-or-flight response, critical to survival, but that has received little attention in humans[22–24]. Of note, this motivational conflict is distinct from "approach-avoidance" dilemmas in which rewards and threats compete for action[25,26], which typically involves a decision about whether one should move closer to danger (e.g., because there might be incentives one can earn[22]). From the perspective of cost-benefit trade-offs and effort-based decision-making, effort-threat motivational conflicts should involve a degree of competition between minimizing effort and maximizing safety, and, in line with the resource-mobilizing and survival-promoting role of the acute stress response[2,4], we predict that its activation should sway decisions in favor of the latter.

To test this hypothesis, 80 participants were randomized to a validated stress-induction procedure that robustly upregulates activation of major stress axes[27] ($n = 40$), or no-stress control condition ($n = 40$), after which they completed a novel task in which they could choose to exert varying amounts of physical effort (here, grip force) in exchange for avoiding varying probabilities of threat-of-shock. To validate this task, we first investigated whether participants explicitly considered effort cost and threat-of-shock in their decision. Next, we applied conventional statistical models and computational cognitive models to compare groups on subjective value computations that underlie the willingness to exert effort in exchange for safety, as well as vigor (i.e., force intensity) at time of effort exertion. Importantly, recent work suggests that induction of acute stress can have mixed, sometimes opposite, effects on the willingness to exert effort in a range of (non-threat) tasks[12–14], suggesting that such effects may be situationally-specific. To investigate whether any potential impact of acute stress on effort-threat motivational conflicts reflects a specific, or general, change in the willingness to exert effort, a second study involving 84 participants was conducted. These participants completed a task in which they could choose to exert physical effort in exchange for monetary rewards[28], thus enabling us to contrast the effect of acute stress on appetitively and aversively motivated behavior. By incorporating effort-based decision-making into studies of threat and acute stress, we aimed to reveal how the acute stress response impacts motivational processes that promote vigorous actions under threat.

## Methods
### Participants
Healthy human participants that decided to take part in Experiment 1 or 2 were screened for eligibility using self-report questions about age (16–35), absence of a DSM (5th ed.)[29] psychiatric disorder, neurological disorder, diabetes type 1 or 2, endocrine and/or cardiovascular disorder (all lifetime), medical implants (current), psychotropic medication use (current), alcohol consumption (<10 units per week), smoking (<10 cigarettes per week), recent substance use (<2 times, past month), very low (<17) or high (>30) body-mass index (current), and pregnancy (current). All participants were informed to not consume any caffeine or food (<2 h before session), alcohol (day of session), or painkillers (day of session, Experiment 1 only), which was confirmed verbally before the start of the study session.

In Experiment 1, data from eight participants were excluded or data were unavailable because of early termination of the experiment ($n = 2$; one at the start of the session and one at the start of stress-induction procedures), failure to adhere to study guidelines ($n = 1$; consumption of coffee and painkillers), a hardware issue ($n = 1$; data not saved), BMI ($n = 1$; BMI ~ 15),

and "inflexible responder" status[14,30,31] (i.e., no variation in choices for all effort-threat combinations; 2 MAST$_{PLC}$, 1 MAST$_{EXP}$), although their inclusion did not change any key results (see below). The final Experiment 1 sample consisted of 80 participants, of which 40 participants were randomized to the Maastricht Acute Stress Task no-stress "placebo" condition (i.e., MAST$_{PLC}$; 27F/13M, age $M = 21.10$, SD = 2.64) and 40 participants to the MAST acute stress "experimental" condition (i.e., MAST$_{EXP}$; 30F/10M, age $M = 21.93$, SD = 2.70).

In Experiment 2, two participants decided to terminate participation early ($n = 1$ before, $n = 1$ during MAST$_{EXP}$ procedures). One additional participant was excluded because they became unwell during the study, and one participant's data were partially missing because of a hardware (saving) issue. The final Experiment 2 sample included 84 individuals, of which 42 were randomized to the MAST$_{PLC}$ condition (32F/10M, age $M = 23.74$, SD = 3.09) and 42 to the MAST$_{EXP}$ condition (31F/11M, age $M = 22.14$, SD = 3.19). Of this sample, four participants completed a version of the reward-effort paradigm discussed below with slightly lower physical effort levels (~5% maximum grip force).

Both studies were approved by the Maastricht University Faculty of Psychology and Neuroscience ethics committee (OZL_242_127_09_2021_ 2022; 220_37_03_2020). All participants provided written informed consent prior to participation. Experiment 1 participants were remunerated in €20 gift vouchers or research participation credits (2 h). Participants in Experiment 2 were remunerated in €15 gift vouchers or research participation credits (1.5 h) plus any task earnings that were, unbeknownst to participants, rounded up to a €5 or €10 gift voucher at the end of the experiment. *A priori* sample size determinations were based on the ability to ensure sufficient power to detect more general medium effects of acute stress on the willingness to exert effort within and between experiments, although the power to detect more complex higher-order interactions with a similar effect size was reduced (e.g., with total $n = 160$: power = 0.88 to detect an Experiment × Condition interaction on choice behavior with effect size $\eta^2 = 0.065$ at alpha = 0.05; power = 0.59 to detect a Condition × Effort × Threat interaction on choice behavior).

### Acute stress induction
The MAST is a psychophysiological stress-induction paradigm that robustly activates autonomic and glucocorticoid systems that mediate the acute stress response[12,27]. The MAST$_{EXP}$ involves negative evaluative feedback from a trained experimenter during a fast-paced backwards counting task, alternated with cold-water immersion of the non-dominant hand (at ~2 °C) while participants continuously view a computer screen showing a (mock) recording of their face. In contrast, the MAST$_{PLC}$ involves simple mental calculations (counting from 1–25 without evaluative performance, no recordings) and room-temperature water. During an initial 5-min introduction phase, participants receive verbal and written instructions about the upcoming procedures, followed by a 10-min stress-induction/control phase during which participants complete alternating blocks of counting and water immersion.

Psychological and physiological stress measurements were collected at multiple timepoints (note: all timepoints mentioned below are relative to the end of MAST/start of the effort-based decision-making paradigm at $t = 0$). In both experiments, saliva samples were obtained at seven timepoints. Two baseline samples were collected prior to starting the MAST control/stress-induction procedures; one at $t = -40$ (min) and one at $t = -10$, with MAST instructions shared 5 min before the $t = -10$ sample and the MAST procedures starting immediately after participants had provided the $t = -10$ sample. The remaining five samples were collected in 10-min intervals, starting immediately post-MAST/at start of the effort-based decision-making paradigm (i.e., $t = 0$ until $t = +40$). All samples were collected using Salivette® swabs (Sarstedt, Etten-Leur, the Netherlands) during a 3-min. sampling period and stored at $-20$ °C until analysis. Salivary cortisol (sCORT) levels, as a proxy of HPA axis activation, were determined using luminescence immune assay kits (IBL, Hamburg, Germany) for all samples. As a proxy measure of sympathetic adrenal axis (SAM) activation,

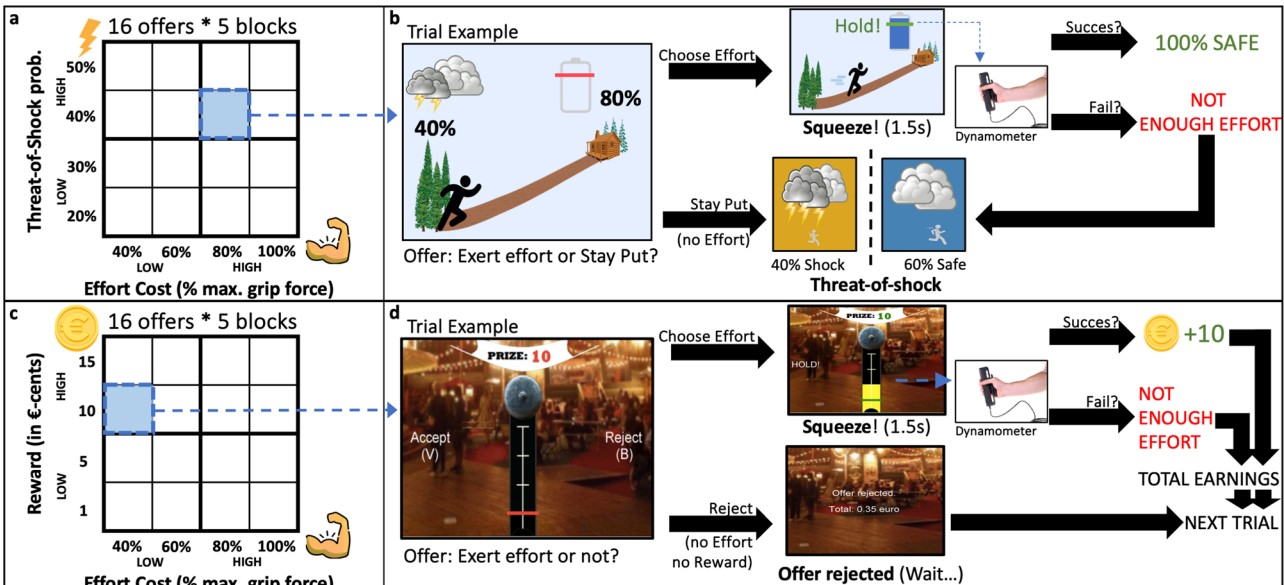

**Fig. 1 | Experiment 1 and 2 paradigm overview.** Graphical overview of the forced-choice threat-effort ("Thunderstorm", Experiment 1) and reward-effort ("High Striker", Experiment 2) paradigms. **a** In the "Thunderstorm" paradigm, participants were presented with 80 trials (16 unique effort cost versus threat-of-shock offers repeated in random order 5 times) and chose between exerting effort (to neutralize the threat-of-shock) or accepting the threat-of-shock (thereby avoiding having to exert effort). **b** Choosing effort was followed by a ~5 s timeframe during which participants were asked to exert above-threshold force for 1.5 consecutive seconds, while acceptance of the threat-of-shock was followed by a ~5 s shock outcome phase according to shock probability. Failure to exert sufficient effort also resulted in the threat-of-shock outcome phase. **c** In the "High Striker" paradigm, participants were presented with 80 trials (16 unique effort-reward offers repeated in random order 5 times) and accepted or rejected the possibility to exert effort in exchange for the reward on offer. **d** Choosing effort was followed by a ~5 s timeframe during which participants were asked to exert above-threshold force for 1.5 consecutive seconds, while rejection of effort was followed by a ~5 s waiting period. Failure to exert sufficient effort did not result in a reward.

salivary alpha-amylase (sAA) levels were—in light of the rapid dynamics of SAM activation[12,27,32]—determined for the first four samples only using a kinetic reaction assay (Salimetrics, Penn State, PA, USA). As additional measures of autonomic reactivity, systolic blood pressure, diastolic blood pressure, and heart rate were measured at $t = -10$ and $t = 0$ using an OMRON M4-I blood pressure monitor (OMRON Healthcare Europe B.V., Hoofddorp, The Netherlands). Pre-post changes in affect were assessed using the Positive and Negative Affect Scale (PANAS; 20-item version)[33] at $t = -10$ and $t = 0$. Study sessions took place between 11:30–17:00 h to minimize effects of diurnal variation in cortisol levels[34].

**Metabolic standardization**
Acute stress, as well as exertion of effort, is associated with marked peripheral autonomic activation[35–37]. Given our primary aim of investigating (cognitive) threat-effort and reward-effort computations impacted by acute stress, we aimed to eliminate large inter-individual differences in energy availability at the start of the experiment. This is especially important because participants are asked to fast as part of MAST procedures, which may additionally impact energy availability and reactivity to stress-induction paradigms[38]. Therefore, in line with previous work[36], all participants consumed a 75 g dextrose chamomile-flavored tea approximately 30 min. prior to the start of the MAST.

**Experiment 1: effort-threat "Thunderstorm" paradigm**
To address the key hypotheses outlined above, we developed an effort-based decision-making paradigm (Fig. 1a, b). Similar to previous work[17–19,25,26,39], and as demonstrated below, the forced-choice nature of this paradigm elicits a motivational conflict, although crucially, here we ask participants to choose between exerting physical effort (i.e., 40–60–80–100% of a participant's pre-calibrated maximum sustainable grip force; MGF) and the possibility of experiencing an aversive electrical shock (i.e., 20–30–40–50% threat-of-shock) (Fig. 1a). By nature of its design, the decision to exert effort can be interpreted as the willingness to exert effort to neutralize the threat-of-shock.

Prior to receiving any task-related information, participants first provided an estimate of their MGF. Participants used their dominant hand to exert maximum force on a handgrip device (i.e., dynamometer; TSD121B-MRI, BIOPAC Systems) during three signaled 5 s calibration trials. Any non-squeeze (baseline) data were removed and remaining data above the median and below 2*standard deviation + median were averaged. In pilot sessions we confirmed that this procedure resulted in an estimate of "sustainable" MGF that participants could exert over multiple trials and for multiple seconds at a time, which is essential given the possibility of repeatedly experiencing painful outcomes.

Next, we obtained a measure of electric shock tolerability using ascending staircasing[40,41]. Over a maximum of 15 trials, participants were asked to rate the intolerability (0–100 scale; dental pain requiring local anesthesia as upper anchor) of a 2 ms electric shock (range: 0–200 V in steps of 20 V) that was delivered via an electric stimulation module (STM100C, BIOPAC Systems) and two gelled electrodes connected to the right ankle (location: adjacent to the *peroneus longus*). Following three (habituation) electric stimulations, the calibration routine would end when participants rated the current electric stimulation as 55–65% intolerable. Calibrated MGF and shock tolerability intensity were subsequently used in the main task to elicit a motivational conflict between minimizing effort and maximizing safety, respectively.

Following initial calibration steps, participants were familiarized with all task procedures during a tutorial session and four practice trials. On every trial, they would see a threat-of-shock (represented as a storm cloud and shock percentage) and the effort cost necessary to neutralize the threat-of-shock (percentage MGF represented as a battery with required force level, and as a hill that their avatar must climb to reach shelter). Next, participants decided if the threat-of-shock, delivered with pre-calibrated intensity, was worth neutralizing by exerting the force level on offer, or—if the threat-of-shock was deemed too low or the force level too high—choose to not exert effort (i.e., stay put), thereby accepting the possibility that the threat may materialize into the experience of pain (Fig. 1b).

Upon choosing effort, participants were required to consecutively exert 1.5 s of above-threshold force (relative to MGF) within a 5 s time window to completely neutralize the threat-of-shock. Force levels were visualized in real time, in combination with messages informing them whether they were exerting sufficient ("HOLD!") or insufficient ("SQUEEZE!") grip force (Fig. 1b). If, at any point, exerted force fell below the required force level, the avatar would restart at the original position. If participants chose to stay put, they would proceed to a threat-of-shock screen represented by a flashing screen and thundercloud, during which the shock outcome was decided and potentially administered. If the participant chose to exert effort, but failed to exert sufficient force, they would transition to the threat-of-shock stage. To control for opportunity cost, effort and threat-of-stock trial stages were closely matched in duration (~10 s for accept effort versus ~9 s for accept threat-of-shock trial stages). During the practice stage, all participants completed a high threat-of-shock/low effort, high threat-of-shock/high effort, low threat-of-shock/low effort, and low threat-of-shock/high effort trial (Fig. 1a).

Immediately post-MAST, participants started the main task during which they completed 80 trials of the forced choice paradigm described above. The above-mentioned four unique levels of effort cost and four unique levels of threat-of-shock were combined into 16 unique trials that were randomly presented and repeated 5 times (i.e., 5 trial "blocks"). Mandatory breaks (~1 min. after every trial block) were introduced to standardize the accumulation of fatigue and cortisol sampling procedures. With a total duration of approximately 27 min., the main task was designed to be completed within the acute stages of a stress response[2].

## Experiment 2: reward-effort "High Striker" paradigm

Participants in Experiment 2 completed a well-validated paradigm in which they could exert varying levels of physical effort in exchange for one of four levels of reward (Fig. 1c). This paradigm is an adapted, shorter, version based on Le Heron et al.[28], that, with a duration of ~23 min, can be completed during the acute stages of a stress response (i.e., prior to recovery), while also allowing for time to obtain saliva samples.

Similar to Experiment 1, an estimate of sustainable MGF was obtained prior to sharing any other task-related information. Next, participants completed four high/low reward/effort practice trials. On each trial, participants were presented with a reward (1, 5, 10, or 15 Eurocents; indicated on a banner at the top of the screen) that could be earned in exchange for a level of physical effort (MGF levels identical to Experiment 1; represented as the red horizontal bar on a high-striker). The self-timed decision to exert effort was followed by a 5 s interval during which participants attempted to complete the trial via 1.5 s of consecutive above-threshold force (visual feedback provided). If participants exerted sufficient force, the reward amount/effort threshold would turn green in combination with a "HOLD!" message. If, at any point, participants exerted insufficient effort, the invisible 1.5 s timer to completion would restart, indicated by the reward amount/ effort threshold turning back to red, in combination with a "SQUEEZE!" message (Fig. 1d).

Exerting sufficient effort was followed by reward feedback ("You won X cents!"), while failure to exert sufficient force was followed by a low-frequency tone and negative feedback ("Failed!"), and rejection of the offer was followed by a time-matched waiting period while "Offer rejected." was displayed on the screen (Fig. 1d). Task earnings were presented briefly in between trials, or during the offer rejected/waiting period. Calibration and practice trials were completed prior to MAST procedures while the main task, involving 80 trials (16 unique effort-reward combinations repeated five times in random order), was completed immediately post-MAST. Participants took mandatory breaks in between each trial block to standardize accumulation of fatigue and cortisol sampling.

## Computational models of effort discounting

Based on Experiment 1 results discussed below, we fit four well-validated computational models of effort discounting to participants' trial-wise choice data to uncover the latent cognitive mechanisms underlying decisions in the

"Thunderstorm" paradigm. All candidate models differed in the exact underlying subjective value computations, which may be linear ($SV = \text{Threat}_{(t)} - k * \text{Effort}_{(t)}$), parabolic ($SV = \text{Threat}_{(t)} - k * \text{Effort}_{(t)}^2$), hyperbolic ($SV = \text{Threat}_{(t)} - (1 / 1 - k * \text{Effort}_{(t)})$) or exponential ($SV = \text{Threat}_{(t)} - e^{k*\text{Effort}(t)}$) in nature, where $k$ is a free subject-level parameter that influences the degree to which effort discounts the value of avoiding a threat[18,19,39]. In line with recent work investigating the trade-off between pain and cognitive effort[39], all models additionally contained a softmax function that determines how subjective value influences choice ($1/ 1 + e^{\beta*(c + SV)}$), where free parameters $\beta$ and $c$ capture the sensitivity to changes in subjective value and a more general choice bias to make one decision over another, respectively.

Effort cost and threat-of-shock were first rescaled to levels (i.e., 1–4). Next, all four models were fit according to a Bayesian two-step hierarchical fitting procedure in which the average and covariance matrix of parameters obtained in the first fitting run were used to shrink the parameter search space in the second run[12,42]. To not bias the fitting procedures towards group differences (in models, or parameters), the mean and covariance matrix were based on the entire sample. Best-fitting subject-level parameters were obtained using the fmincon function in MATLAB (v.2021b; Mathworks, Natick, MA, USA), and the best-fitting model for each group was identified using Bayesian Model Selection[43] with Akaike Information Criterion (AIC)[44] as the fit metric (spm_BMS function in SPM12, http://www.fil.ion. ucl.ac.uk/spm/software/spm12/). Model-predicted subjective value estimates from the most likely model were extracted and compared between groups in multilevel regression models specified below.

## Statistical analyses

For each experiment, condition (MAST_PLC, MAST_EXP; IV) differences in demographics and self-rated questionnaire scores (DV) were assessed using independent samples $t$-tests or Chi square tests, where applicable. To confirm that stress-induction was successful, conditions were moreover compared on pre-post differences in subjective, autonomic, and glucocorticoid stress measurements using repeated-measures ANOVAs (i.e., Condition × Time interactions), followed by *post hoc* independent samples and paired-samples $t$-tests. When using sAA data as the dependent variable, pre- and post-MAST samples were averaged separately in light of substantial variability in SAM activation. Relationships between calibration variables (MGF in both experiments, shock intensity in Experiment 1) and the proportion of trials on which participants chose to exert effort were investigated using Pearson's $r$ correlation analyses.

Next, multilevel logistic regression models were used to assess main and interaction effects with choice as the dependent variable (accept effort = 1, accept threat-of-shock in Experiment 1/forego reward in Experiment 2 = 0; data structure: trial-level data for every participant). First, to validate the Thunderstorm paradigm, we evaluated main effects of effort cost (i.e., offer MGF %, 4 levels), threat-of-shock (i.e., offer shock probability, 4 levels), and time (i.e., trial block, 5 levels) on choice behavior, expecting that effort cost (negatively) and threat-of-shock (positively), but not time (due to a sustainable level of MGF), would be associated with decisions to exert effort in exchange for safety. In this model, all three predictors were added as random intercept and slopes at the subject level. This model structure was also used when evaluating the effect of task variables on choice behavior in Experiment 2, with the obvious exception that threat-of-shock was replaced with reward value (i.e., Eurocents on offer, 4 levels).

Secondly, to investigate if acute stress increases the willingness to exert effort to avoid threats, we assessed the presence of a (MAST) Condition × Effort cost × Threat-of-shock interaction in the model of choice, with threat-of-shock and effort cost as fixed and random effects at the subject level. Condition was added as fixed and random effect at the subject level to allow the intercept variance to differ across conditions[45,46]. This multilevel model structure was used for all other GLMs that assessed condition differences, including the comparison of model-predicted subjective value estimates between conditions in Experiment 1 (using a linear regression) and Experiment 2 analyses (with threat-of-shock replaced by reward value).

When investigating associations between choice data and MAST-induced affect, autonomic, and glucocorticoid changes (i.e., post minus pre differences scores), the same multilevel model structure was used, and stress predictors were only added as random intercept at the subject level. For these analyses, Bonferroni correction was applied for the number of stress parameters in the model. Observed (Condition × Effort cost × Threat-of-shock) interactions in Experiment 1 were followed up by a *post hoc* test assessing the presence of an Effort Cost × Threat-of-shock interaction for each condition separately. To further break down this interaction, we assessed condition differences for four distinct offer types (i.e., high/low effort/threat; Fig. 1a). This approach allowed us to uncover condition differences that were observable across a reasonable part of the offer sampling space (i.e., 25%), thereby limiting the number of tests, and thus the false positive rate.

In Experiment 1, condition differences in exerted force intensity following the decision to exert effort, a proxy measure of aversive vigor[23], were also investigated. For each trial, we identified a 2.5 s epoch of data during which participants were successfully ramping up and maintaining force to neutralize the threat-of-shock. To minimize the influence of extreme values, a 100 ms rolling average was applied to the force data, which were originally sampled at 60 Hz. Two metrics that capture how vigorously participants exerted effort to avoid threats were calculated. First, we calculated average overexertion, i.e., the amount of force (in % MGF) by which participants exceeded the required force level. Secondly, we obtained the positive "yank peak"—the maximal first derivative of force—for each trial, which represents a measure of change in force applied[47]. In force data analyses we were unable to investigate the presence of higher-order interactions (e.g., Condition × Effort Cost × Threat-of-shock) with regression models containing multiple subject-level random slopes. Estimating these models resulted in singularity and estimation issues, likely due to extremely limited availability of force data for effort-threat combinations with negative objective value. We therefore investigated only a main effect of condition, taking into account effort cost (both as random intercept and slope at the subject level), although highly similar results were obtained when accounting for threat-of-shock and block in separate analyses (included in accompanying code).

Multilevel regression analyses were conducted in R (v4.3.1) using the lme4 package (glmer for binomial choice data; lmer for continuous subjective value estimates and force data[46]; quadratic approximation function BOBYQA[48]). Effort cost threat-of-shock, and reward levels were rescored to 1–4 (i.e., assuming linear increases) and zero-centered. Linear hypothesis testing (linearHypothesis in car package[49]) and stratified analyses were used to conduct *post hoc* comparisons, where applicable. To ensure stability of the observed results, all reported 95% confidence intervals were bootstrapped (500 iterations, confint in MASS package[50] for glm functionality). For Experiment 2, key GLMs were re-run in a Bayesian analysis framework to quantify the evidence for an absence of condition-related effects (brm and bayes_factor functions in brms package[51]; 20,000 iterations). The statistical analyses and models reported above were not pre-registered. Unless explicitly stated otherwise, all reported results are based on the entire sample of $n = 80$ (for Experiment 1 analyses) and $n = 84$ (for Experiment 2 analyses).

## Reporting summary
Further information on research design is available in the Nature Portfolio Reporting Summary linked to this article.

## Results
### Experiment 1 sample characteristics
All demographic and self-report questionnaire scores and test statistics for Experiment 1 $MAST_{PLC}$ and $MAST_{EXP}$ groups are summarized in Supplementary Table 1. Experiment 1 participants randomly assigned to $MAST_{PLC}$ and $MAST_{EXP}$ did not significantly differ (at $p = 0.05$) on maximum grip force, shock intensity tolerance, age, sex, anticonception use, menstrual phase, and various self-rated measures of motivation, chronic stress, avoidance, and fear of pain, except for a subtle difference in the social

motivation subscale of the Motivation and Energy Inventory (MEI)[52] ($MAST_{PLC} > MAST_{EXP}$; Supplementary Table 1).

### Experiment 1 acute stress manipulation
We next investigated if the MAST protocol successfully elicited an affective, autonomic, and glucocorticoid acute stress response in Experiment 1 participants.

Pre-MAST, $MAST_{EXP}$ and $MAST_{PLC}$ groups did not significantly differ on negative affect ($t_{(78)} = 1.73$, $p = 0.09$, $d = 0.39$, 95% CI = −0.03–0.84), positive affect ($t_{(78)} = 1.24$, $p = 0.22$, $d = 0.28$, 95% CI = −0.17–0.76), systolic blood pressure ($t_{(78)} = 0.59$, $p = 0.56$, $d = 0.13$, 95% CI = −0.30–0.60), diastolic blood pressure ($t_{(78)} = 0.53$, $p = 0.60$, $d = 0.12$, 95% CI = −0.32–0.59), heart rate ($t_{(78)} = 0.14$, $p = 0.89$, $d = 0.04$, 95% CI = −0.41–0.52), sCORT ($t_{(78)} = 0.09$, $p = 0.93$, $d = 0.02$, 95% CI = −0.42–0.47) and sAA ($t_{(78)} = −0.46$, $p = 0.65$, $d = −0.10$, 95% CI = −0.66–0.36). However, significant Condition × Time interactions were observed for affect (negative: $F_{(1,78)} = 56.43$, $p < 0.001$, $\eta^2_G = 0.14$, 95% CI = 0.03–0.27; positive: $F_{(1,78)} = 10.65$, $p < 0.01$, $\eta^2_G = 0.02$, 95% CI = 0.00–0.07), systolic blood pressure ($F_{(1,78)} = 67.93$, $p < 0.001$, $\eta^2_G = 0.13$, 95% CI = 0.02–0.26), diastolic blood pressure ($F_{(1,78)} = 93.25$, $p < 0.001$, $\eta^2_G = 0.22$, 95% CI = 0.08–0.36), heart rate ($F_{(1,78)} = 12.13$, $p < 0.001$, $\eta^2_G = 0.02$, 95% CI = 0.00–0.07), and sCORT ($F_{(5,390)} = 14.67$, $p < 0.001$, $\eta^2_G = 0.05$, 95% CI = 0.00–0.08) (Fig. 2). We did not observe evidence for a Condition × Time interaction for sAA ($F_{(1,77)} = 0.81$, $p = 0.37$, $\eta^2_G < 0.01$, 95% CI = 0.00–0.03; $n = 1$ missing data), although $MAST_{EXP}$ participants ($t_{(39)} = 2.00$, $p = 0.05$, $d = 0.32$, 95% CI = 0.04–0.56), but not $MAST_{PLC}$ participants ($t_{(38)} = 1.23$, $p = 0.23$, $d = 0.20$, 95% CI = −0.15–0.56), exhibited a marginally significant pre-to-post increase in sAA.

Post hoc simple main effect analyses revealed that only the $MAST_{EXP}$ group exhibited significant pre-to-post-MAST increases in negative affect ($MAST_{EXP}$ post/pre: $t_{(39)} = 5.46$, $p < 0.001$, $d = 0.86$, 95% CI = 0.63–1.19; $MAST_{PLC}$ post/pre: $t_{(39)} = −6.38$, $p < 0.001$; $d = −1.01$, 95% CI = −1.44 to −0.82; $MAST_{PLC}$-$MAST_{EXP}$ post: $t_{(78)} = −5.17$, $p < 0.001$, $d = −1.16$, 95% CI = −1.69 to −0.71), systolic blood pressure ($MAST_{EXP}$ post/pre: $t_{(39)} = 9.96$, $p < 0.001$, $d = 1.57$, 95% CI = 1.13–2.28; $MAST_{PLC}$ post/pre: $t_{(39)} = −0.24$, $p = 0.81$, $d = −0.04$, 95% CI = −0.38–0.24; $MAST_{PLC}$-$MAST_{EXP}$ post: $t_{(78)} = −5.98$, $p < 0.001$, $d = −1.34$, 95% CI = −1.96 to −0.86), diastolic blood pressure ($MAST_{EXP}$ post/pre: $t_{(39)} = 11.86$, $p < 0.001$, $d = 1.88$, 95% CI = 1.49–2.47; $MAST_{PLC}$ post/pre: $t_{(39)} = −0.85$, $p = 0.40$, $d = −0.13$, 95% CI = −0.39–0.20; $MAST_{PLC}$-$MAST_{EXP}$ post: $t_{(78)} = −7.84$, $p < 0.001$, $d = −1.75$, 95% CI = −2.28 to −1.36), heart rate ($MAST_{EXP}$ post/pre: $t_{(39)} = 2.80$, $p < 0.01$, $d = 0.44$, 95% CI = 0.12–0.90; $MAST_{PLC}$ post/pre: $t_{(39)} = −2.13$, $p = 0.04$, $d = −0.34$, 95% CI = −0.75 to −0.01; $MAST_{PLC}$-$MAST_{EXP}$ post: $t_{(78)} = −2.09$, $p = 0.04$, $d = −0.47$, 95% CI = −0.95 to −0.06), and greater positive affect decreases than the $MAST_{PLC}$ group ($MAST_{EXP}$ post/pre: $t_{(39)} = −6.09$, $p < 0.001$, $d = −0.96$, 95% CI = −1.56 to −0.52; $MAST_{PLC}$ post/pre: $t_{(39)} = −2.68$, $p = 0.01$, $d = −0.42$, 95% CI = −0.75 to −0.12; $MAST_{PLC}$-$MAST_{EXP}$ post: $t_{(78)} = 3.32$, $p < 0.01$, $d = 0.74$, 95% CI = 0.33–1.23). Moreover, $MAST_{EXP}$ versus $MAST_{PLC}$ participants exhibited greater peak sCORT levels ($t_{(78)} = 5.01$, $p < 0.001$, $d = 1.12$, 95% CI = 0.82–1.55) and greater sCORT levels at all post-MAST timepoints (available in accompanying code) (Fig. 2).

All in all, these results confirm that the MAST robustly increased affective, autonomic, and glucocorticoid indices of acute stress in Experiment 1 participants.

### Forced choices between avoiding effort versus threats elicit a motivational conflict
We conducted several diagnostic checks to confirm that the paradigm employed in Experiment 1 worked as intended. First, we found no evidence for a correlation between decisions to exert effort and calibrated shock intensity ($MAST_{PLC}$: $r_{(38)} = 0.04$, 95% CI = −0.28–0.35, $p = 0.81$; $MAST_{EXP}$: $r_{(38)} = 0.01$, 95% CI = −0.30–0.32, $p = 0.95$) or calibrated maximum sustained grip force (MGF) ($MAST_{PLC}$: $r_{(38)} = −0.25$, 95% CI = −0.52–0.06, $p = 0.11$; $MAST_{EXP}$: $r_{(38)} = −0.22$, 95% CI = −0.50–0.10, $p = 0.17$).

**Fig. 2 | Experiment 1 acute stress-induction results.** Group differences in affective, autonomic, and glucocorticoid markers of acute stress for Experiment 1 participants. Timepoints ($t$) are relative to end of MAST/start of the "Thunderstorm" paradigm. $p \leq 0.05$, $**p \leq 0.01$, $***p \leq 0.001$. Error bars represent SD. $n = 80$ participants.

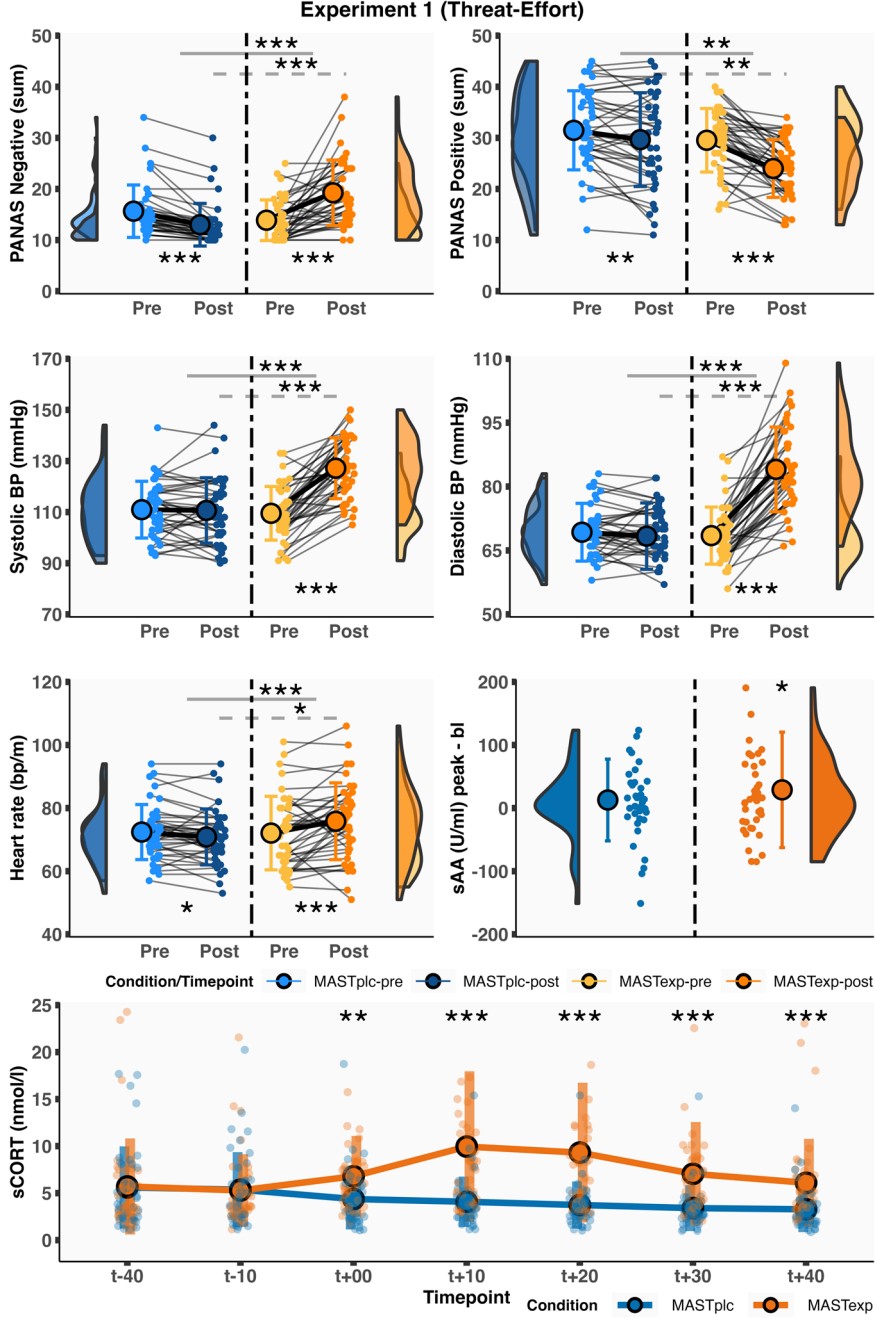

Secondly, MAST$_{PLC}$ and MAST$_{EXP}$ participants did not significantly differ in the amount of failed (effort) trials when considering all trials ($t_{(78)} = 0.11$, $p = 0.92$, $d = 0.03$, 95% CI = $-0.44$–$0.46$) or only the most effortful trials ($t_{(78)} = -1.50$, $p = 0.14$, $d = -0.35$, 95% CI = $-0.79$–$0.08$). Third, multilevel logistic regression analyses revealed that effort cost ($B_{EFFORT} = -1.47$, 95% CI = $-1.73$ to $-1.22$, $p < 0.001$), threat-of-shock probability ($B_{THREAT} = 2.21$, 95% CI = $1.96$–$2.48$, $p < 0.001$) (Fig. 3a), but not time ($B_{BLOCK} = -0.09$, 95% CI = $-0.21$–$0.02$, $p = 0.11$), were significantly associated with decisions to exert effort. Overall, these results suggest that this paradigm has the potential to elicit motivational conflicts between minimizing effort and maximizing safety, in the absence of a marked influence of, and group differences in, calibration values and/or failure rates.

**Acute stress increases the willingness to exert effort in the service of avoiding threats**

We next used Experiment 1 data to investigate how the acute stress response impacts the willingness to exert effort in the service of avoiding threats.

Importantly, a (MAST) Condition × Effort Cost × Threat-of-shock interaction on choice ($B_{CxExT} = -0.21$, 95% CI = $-0.40$–$0.00$, $p = 0.029$) was observed, which remained significant when correcting for calibrated shock intensity and maximum sustained grip force ($B_{CxExT} = -0.20$, 95% CI = $-0.43$–$0.00$, $p = 0.036$), and when including participants that chose to exert effort on every trial ("inflexible responders"; $B_{CxExT} = -0.21$, 95% CI = $-0.44$ to $-0.01$, $p = 0.031$). This interaction was driven by a significant Effort Cost × Threat-of-shock interaction in the MAST$_{EXP}$ ($p < 0.001$, $\chi^2 = 17.89$), but not MAST$_{PLC}$ ($p = 0.10$, $\chi^2 = 2.75$), group.

In *post hoc* analyses of four distinct offer types (see Fig. 1a for quadrants), we observed that MAST$_{EXP}$ versus MAST$_{PLC}$ participants exhibited a selective increase in the tendency to choose effort on low threat-of-shock/high effort trials ($B_{CONDITION} = 1.44$, 95% CI = $0.44$–$2.57$, $p = 0.01$), but not high threat/low effort ($B_{CONDITION} = 0.10$, 95% CI = $-2.92$–$1.68$, $p = 0.90$), high threat/high effort ($B_{CONDITION} = 0.99$, 95% CI = $-0.34$–$2.23$, $p = 0.09$), or low threat/low effort trials ($B_{CONDITIONP} = 0.72$, 95% CI = $-0.65$–$2.07$, $p = 0.28$). Assuming 1:1 correspondence between effort and threat-of-shock

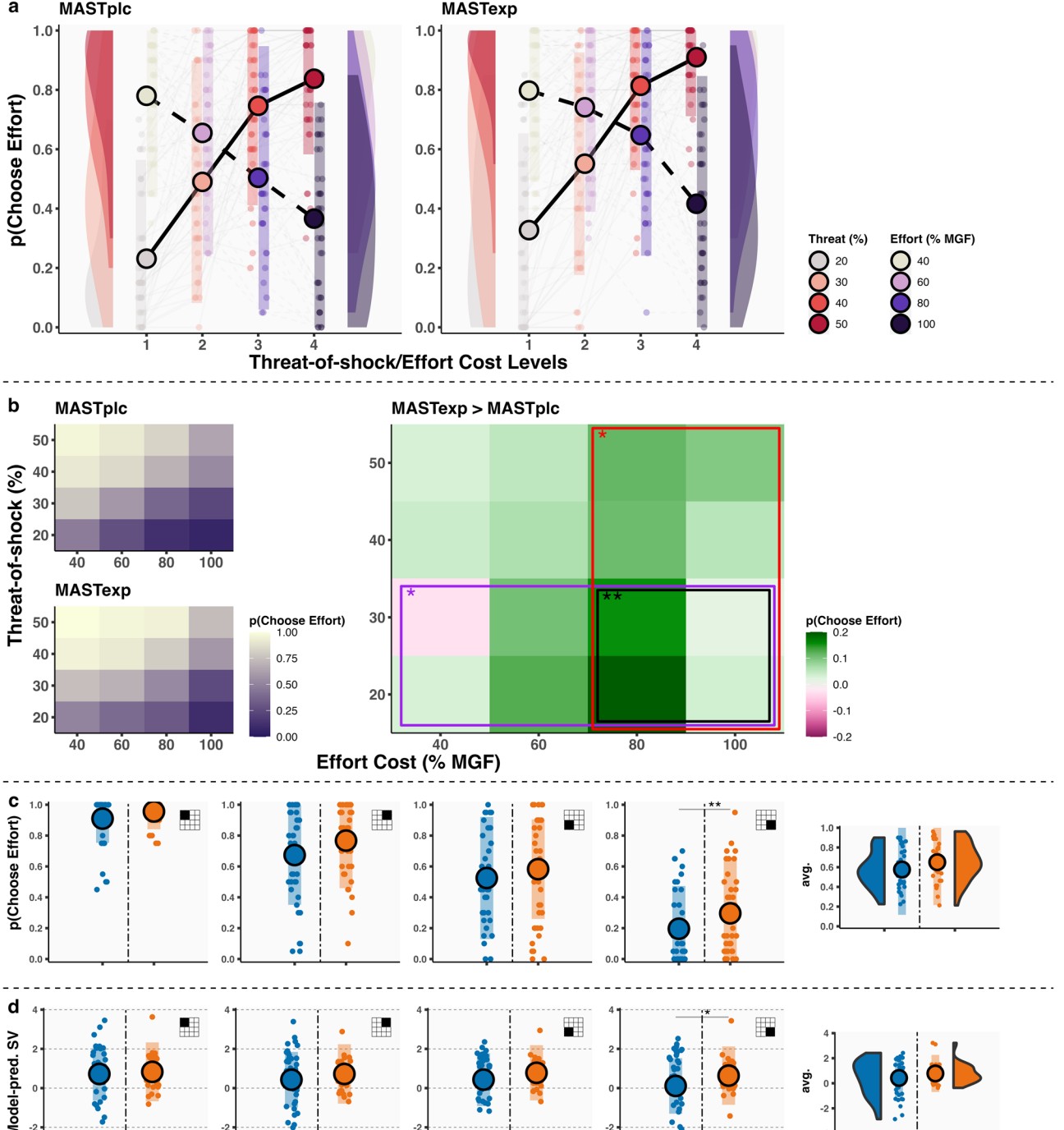

**Fig. 3 | Thunderstorm task choice data and computational modeling results (Experiment 1). a** In Experiment 1, increasing effort cost and threat-of-shock levels (*x*-axis) were negatively/positively associated with a greater likelihood of accepting effort (*y*-axis). **b** Heatmaps of average effort acceptance for the 16 unique offers that can be created by combining the 4 effort cost (*x*-axis) and 4 threat-of-shock (*y*-axis) levels. Shown for MAST$_{PLC}$ and MAST$_{EXP}$ participants separately (left, in white-to-purple) and as a group difference score (right, in green-to-pink). **c** Effort acceptance (*y*-axis) for low/high effort/threat offers (see each plot's inset). **d** Model-predicted subjective value of exerting effort (*y*-axis) for low/high effort/threat offers. Note: all plots are based on raw participant data. *$p \leq 0.05$, **$p \leq 0.01$. Error bars represent SD. $n = 80$ participants.

levels, these results suggest that MAST$_{EXP}$ participants were more likely to choose effort on trials where exerting effort has negative objective value (Fig. 3b, c).

In line with these observations, MAST$_{EXP}$ compared to MAST$_{PLC}$ participants also exhibited a subtle, marginally significant, more general tendency to choose effort on high effort trials ($B_{CONDITION} = 0.96$, 95% CI = 0.09–1.87, $p = 0.046$) and low threat-of-

shock trials ($B_{CONDITIONP} = 1.03$, 95% CI = 0.06–2.02, $p = 0.050$), independent of the other option on offer (Fig. 3b). Effort acceptance rates on low threat/high effort trials in MAST$_{EXP}$ participants were significantly positively associated with MAST-induced changes in negative affect ($B_{NA.DIFF} = 0.14$, 95% CI = 0.03–0.25, $p_{Bonf} = 0.02$), but not with systolic blood pressure ($B_{SYS.DIFF} = 0.05$, 95% CI = −0.01–0.11, $p_{Bonf} = 0.32$) or cortisol ($B_{CORT.DIFF} = 0.06$, 95% CI = −0.02–0.14, $p_{Bonf} = 0.48$).

**Fig. 4 | Thunderstorm task aversive vigor.** Greater force intensity following the decision to exert effort to avoid the threat-of-shock in MAST_EXP compared to MAST_PLC participants; both when considering run-up and maintenance of force (**a**), and after excluding data that may have been part of previous (unsuccessful) attempts to complete the trial (**b**). Note: trial-level raw data (individual data points) and group-level summaries of individual-level averaged raw data (mean + error bars) are both shown. Greater *y*-axis values represent greater above-threshold grip force (i.e., overexertion) in % MGF. *$p \leq 0.05$, **$p \leq 0.01$. Error bars represent SD. *n* = 80 participants.

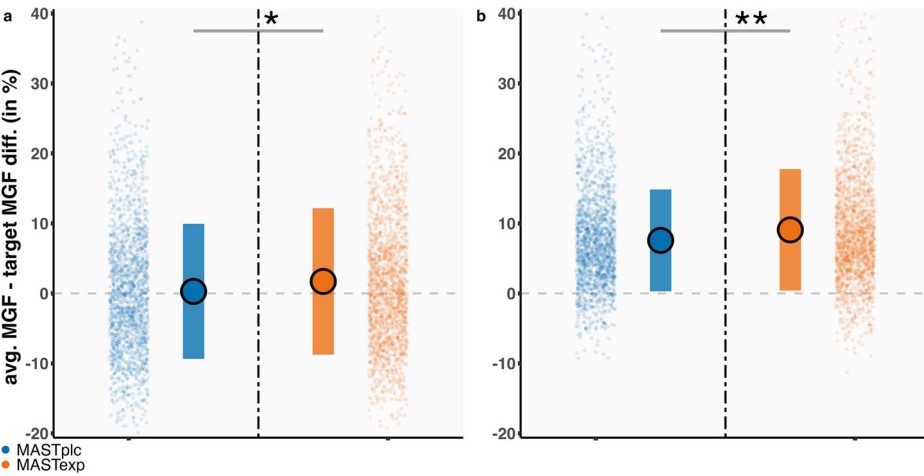

### Acute stress alters integrative effort-threat computations

To obtain mechanistic insights into the impact of acute stress on choice behavior in Experiment 1, we next investigated group differences in the subjective value computations that dictate how effort-threat motivational conflicts are resolved. We fit four computational models of effort discounting to trial-wise data[18,19,39]. Although all models assume that decisions arise from a process involving the subtraction of effort cost from the "value" (i.e., relief) of avoiding a threat, the mathematical formulation controlling the effort-threat trade-off differed per model.

MAST_PLC and MAST_EXP participants were fit best by different models. MAST_PLC choice data were captured best by a linear discounting function (i.e., $\text{Threat}_{(t)} - k*\text{Effort}_{(t)}$; exceedance probability xp = 0.98, pxp = 0.40; expectation of the posterior: p[r | y] = 0.51), which assumes consistent discounting across levels of effort cost. In contrast, the most likely model for the MAST_EXP choice data included a hyperbolic discounting function ($\text{Threat}_{(t)} - (1 / 1 - k*\text{Effort}_{(t)})$; xp = 0.93, pxp = 0.42; p[r | y] = 0.51), which assumes greater reductions in the subjective value of exerting effort for lower versus higher levels of effort cost[18,39]. In line with the empirical choice data, MAST_EXP compared to MAST_PLC participants exhibited significantly higher model-predicted subjective value of exerting effort on low threat-of-shock/high effort trials ($B_{\text{CONDITION}} = 0.68$, 95% CI = 0.05–1.31, $p = 0.028$), but we found no evidence for group differences on high threat-of-shock/low effort ($B_{\text{CONDITION}} = 0.25$, 95% CI = −0.27–0.74, $p = 0.33$), high threat-of-shock/high effort ($B_{\text{CONDITION}} = 0.25$, 95% CI = −0.30–0.78, $p = 0.34$), or low threat-of-shock/low effort trials ($B_{\text{CONDITION}} = 0.28$, 95% CI = −0.19–0.75, $p = 0.21$) (Fig. 3d).

These results suggest that acute stress sways the effort-threat motivational conflict towards maximizing safety by reducing the negative impact of increasing effort cost on the willingness to avoid threats. This, in turn, facilitates effort expenditure in the service of avoiding threats; especially those threats that are unlikely to materialize in the first place.

### Acute stress increases aversive vigor

Although analyses of Experiment 1 choice behavior suggested that MAST_EXP compared to MAST_PLC participants were more willing to engage in vigorous actions aimed at avoiding threats, we also directly investigated the vigor of avoidance actions (i.e., aversive vigor). In epochs of high-resolution (60 Hz) grip force data signifying a successful run-up and maintenance of force to neutralize the threat-of-shock, MAST_EXP compared to MAST_PLC participants exerted more above-threshold grip force to neutralize the threat-of-shock ($B_{\text{CONDITION}} = 2.09$, 95% CI = 0.33–4.18, $p = 0.035$) (Fig. 4a). Group differences in aversive vigor remained significant when considering a shorter (2 s) epoch that was less susceptible to confounding by previous attempts to complete the trial ($B_{\text{CONDITION}} = 1.68$, 95% CI = 0.53–2.98, $p < 0.01$) (Fig. 4b). These results indicate that acute stress facilitates more vigorous application of force in the service of avoiding

threats. Group differences in the rate of change in force applied (over time), also known as yank peak[47], were not observed ($B_{\text{CONDITION}} = -0.05$, 95% CI = 0.77–0.70, $p = 0.89$) (Supplementary Fig. 1).

### Experiment 2 sample characteristics and acute stress manipulation

Having observed that acute stress alters the trade-off between minimizing effort and maximizing safety, we used Experiment 2 data to assess if stress-induced changes in the willingness to exert effort may also be apparent in non-threatening situations. All demographic and self-report questionnaire scores and test statistics for Experiment 2 MAST_PLC and MAST_EXP groups are reported in Supplementary Table 2. Except for a slight difference in age, Experiment 2 MAST_PLC and MAST_EXP groups did not significantly differ on a range of demographic variables, maximum grip force, or self-report measures of motivation, chronic stress, or hedonic capacity (Supplementary Table 2).

At baseline (i.e., pre-MAST), Experiment 2 MAST_EXP and MAST_PLC participants did not significantly differ on measures of negative affect ($t_{(82)} = -0.72$, $p = 0.48$, $d = -0.16$, 95% CI = −0.55–0.32), positive affect ($t_{(82)} = -0.21$, $p = 0.84$, $d = -0.05$, 95% CI = −0.48–0.35), systolic blood pressure ($t_{(82)} = -0.18$, $p = 0.86$, $d = -0.04$, 95% CI = −0.50–0.42), diastolic blood pressure ($t_{(82)} = 1.10$, $p = 0.27$, $d = 0.24$, 95% CI = −0.17–0.68), heart rate ($t_{(82)} = 0.87$, $p = 0.38$, $d = 0.19$, 95% CI = −0.22–0.67), sCORT ($t_{(82)} = 0.21$, $p = 0.83$, $d = 0.05$, 95% CI = −0.43–0.42), and sAA ($t_{(82)} = -0.68$, $p = 0.50$, $d = -0.15$, 95% CI = −0.56–0.28). However, significant Condition × Time interactions were observed for affect ratings (negative: $F_{(1,82)} = 42.77$, $p < 0.001$, $\eta^2_G = 0.09$, 95% CI = 0.00–0.21; positive: $F_{(1,82)} = 4.47$, $p = 0.04$, $\eta^2_G = 0.01$, 95% CI = 0.01–0.04), systolic blood pressure ($F_{(1,82)} = 44.78$, $p < 0.001$, $\eta^2_G = 0.09$, 95% CI = 0.01–0.20), diastolic blood pressure ($F_{(1,82)} = 52.67$, $p < 0.001$, $\eta^2_G = 0.15$, 95% CI = 0.04–0.28), heart rate ($F_{(1,82)} = 13.04$, $p = 0.001$, $\eta^2_G = 0.02$, 95% CI = 0.00–0.08), and sCORT ($F_{(5, 410)} = 19.52$, $p < 0.001$, $\eta^2_G = 0.05$, 95% CI = 0.00–0.08) (Fig. 5).

Simple main effect analyses revealed that only the MAST_EXP group exhibited significant pre-to-post increases in negative affect (MAST_EXP post/pre: $t_{(41)} = 5.87$, $p < 0.001$, $d = 0.91$, 95% CI = 0.69–1.18; MAST_PLC post/pre: $t_{(41)} = -2.93$, $p < 0.01$, $d = -0.45$, 95% CI = −0.86 to −0.16; MAST_PLC-MAST_EXP post: $t_{(82)} = -5.83$, $p < 0.001$, $d = -1.27$, 95% CI = −1.81 to −0.89), systolic blood pressure (MAST_EXP post/pre: $t_{(41)} = 8.34$, $p < 0.001$, $d = 1.29$, 95% CI = 1.05–1.65; MAST_PLC post/pre: $t_{(41)} = -0.50$, $p = 0.62$, $d = -0.08$, 95% CI = −0.31–0.29; MAST_PLC-MAST_EXP post: $t_{(82)} = -5.78$, $p < 0.001$, $d = -1.26$, 95% CI = −1.82 to −0.83), diastolic blood pressure (MAST_EXP post-pre: $t_{(41)} = 8.31$, $p < 0.001$, $d = 1.28$, 95% CI = 0.94–1.84; MAST_PLC post/pre: $t_{(41)} = -0.93$, $p = 0.36$, $d = -0.14$, 95% CI = −0.54–0.15; MAST_PLC-MAST_EXP post: $t_{(82)} = -6.06$, $p < 0.001$, $d = -1.32$, 95% CI = −1.85 to −0.92), and heart rate (MAST_EXP post/pre: $t_{(41)} = 3.51$, $p = 0.001$, $d = 0.54$, 95% CI = 0.27–0.90; MAST_PLC post/pre:

**Fig. 5 | Experiment 2 acute stress-induction results.** Group differences in affective, autonomic, and glucocorticoid markers of acute stress for Experiment 2 participants. Timepoints ($t$) are relative to end of MAST/start of the "High Striker" paradigm. $p \leq 0.05$, $**p \leq 0.01$, $***p \leq 0.001$. Error bars represent SD. $n = 84$ participants.

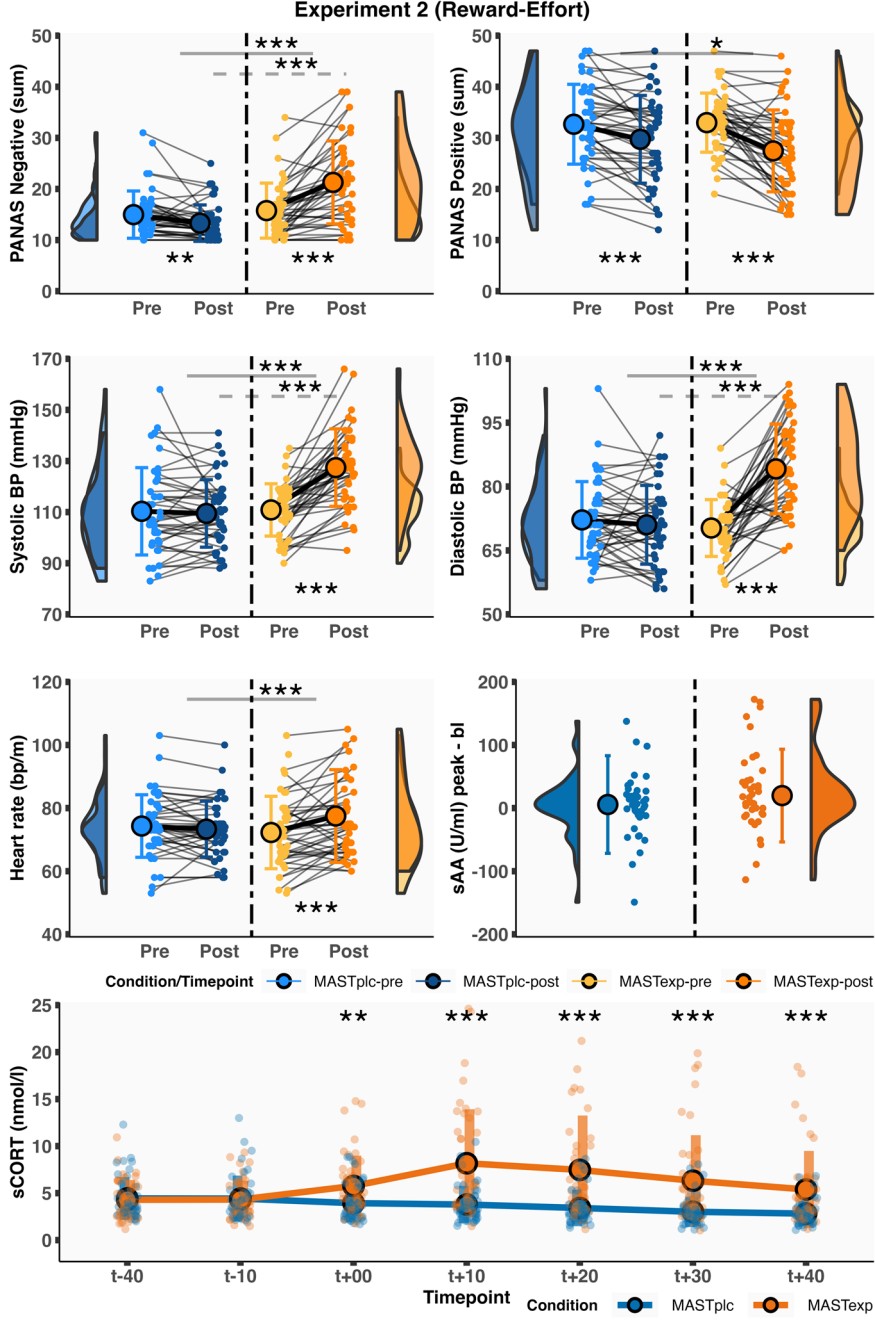

$t_{(41)} = -1.15$, $p = 0.26$, $d = -0.18$, 95% CI = $-0.54$–$0.14$; MAST$_{PLC}$-MAST$_{EXP}$ post: $t_{(82)} = -1.56$, $p = 0.12$, $d = -0.34$, 95% CI = $-0.75$–$0.07$). Both groups exhibited a significant pre-to-post decrease in positive affect (MAST$_{EXP}$ post/pre: $t_{(41)} = -5.83$, $p < 0.001$, $d = -0.90$, 95% CI = $-1.36$ to $-0.55$; MAST$_{PLC}$ post/pre: $t_{(41)} = -3.87$, $p < 0.001$, $d = -0.60$, 95% CI = $-0.91$ to $-0.33$; MAST$_{PLC}$-MAST$_{EXP}$ post-MAST: $t_{(82)} = 1.25$, $p = 0.22$, $d = 0.27$, 95% CI = $-0.13$–$0.75$), suggesting that the interaction was driven by a greater positive affect change in the MAST$_{EXP}$ versus MAST$_{PLC}$ group. MAST$_{EXP}$ compared to MAST$_{PLC}$ participants also exhibited greater baseline-corrected peak sCORT ($t_{(82)} = 5.59$, $p < 0.001$, $d = 1.22$, 95% CI = $0.92$–$1.62$) and greater sCORT levels at all post-MAST timepoints (available in accompanying code) (Fig. 5).

We found no evidence for a significant Condition × Time interaction for sAA ($F_{(1,82)} = 0.74$, $p = 0.39$, $\eta^2_G = 0.001$, 95% CI = $0.00$–$0.02$), likely owing to the extreme variability in sAA measurements in each group. Similar to Experiment 1, MAST$_{EXP}$ participants exhibited a greater

numerical increase in sAA, which failed to reach significance (MAST$_{EXP}$: $t_{(41)} = 1.75$, $p = 0.088$, $d = 0.27$, 95% CI = $-0.05$–$0.64$; MAST$_{PLC}$: $t_{(41)} = 0.47$, $p = 0.64$, $d = 0.07$, 95% CI = $-0.24$–$0.37$). Taken together, these results suggest an overall pattern of group differences in acute stress parameters that is almost identical to Experiment 1.

## No evidence for a stress-induced change in the willingness to exert effort in exchange for rewards

Similar to Experiment 1, we observed no evidence for relationships between calibrated force levels and decisions to exert effort (MAST$_{PLC}$: $r_{(40)} = -0.07$, 95% CI = $-0.36$–$0.24$, $p = 0.68$; MAST$_{EXP}$: $r_{(40)} = -0.16$, 95% CI = $-0.44$–$0.15$, $p = 0.30$) or group differences in failed (effort) trials (all trials: $t_{(82)} = -1.45$, $p = 0.15$, $d = -0.32$, 95% CI = $-0.70$–$0.11$; 100% MGF trials: $t_{(82)} = -0.59$, $p = 0.56$, $d = -0.13$, 95% CI = $-0.56$–$0.30$).

Multilevel logistic regression analyses revealed large main effects of effort cost ($B_{EFFORT} = -2.18$, 95% CI = $-2.52$–$1.85$, $p < 0.001$), reward value

($B_{REWARD}$ = 3.66, 95% CI = 3.13–4.16, $p < 0.001$) (Fig. 6a), but not time ($B_{BLOCK}$ = −0.05, 95% CI = −0.22–0.10, $p = 0.50$), on choice. However, there was no evidence for a Condition × Reward × Effort Cost ($B_{CxRxE}$ = 0.07, 95% CI = −0.43–0.52, $p = 0.74$), Condition × Reward ($B_{CxR}$ = 0.12, 95% CI = −0.88–1.21, $p = 0.83$), Condition × Effort Cost interaction ($B_{CxE}$ = −0.01, 95% CI = −0.86–0.91, $p = 0.98$), or main effect of Condition ($B_{CONDITION}$ = −0.39, 95% CI = −2.09–1.35, $p = 0.67$) on choice (Fig. 6b, c; see Supplementary Table 3 for full model output). Similarly, Bayesian multilevel logistic regression analyses provided no credible evidence for a Condition × Reward × Effort Cost (parameter mean/$param_m$ = 0.12, parameter 95% CI/$param_{95\%}$ = −0.27–0.51), Condition × Reward ($param_m$ = −0.11, $param_{95\%}$ = −1.11–0.90), Condition × Effort Cost interaction ($param_m$ = 0.12, $param_{95\%}$ = −0.67–0.90), or main effect of Condition ($param_m$ = −0.83, $param_{95\%}$ = −2.59–0.95), as demonstrated by credible intervals that all included zero. Directly comparing model evidence of statistical models with(out) a Condition × Reward × Effort term provided strong evidence in favor of the model without this interaction

($BF_{01}$ = 10.66). Moreover, comparing model evidence for models with(out) a main effect of Condition provided anecdotal evidence against a model without this term ($BF_{01}$ = 0.35, i.e., $BF_{10}$ = 2.86). However, here, it is noteworthy that, numerically, the main effect of Condition in Experiment 2 suggested that $MAST_{EXP}$ compared to $MAST_{PLC}$ participants were less likely to exert effort in exchange for reward.

Finally, we directly compared the effect of acute stress on the willingness to exert effort in both experiments, finding a significant Experiment × Condition interaction ($B_{EXPxC}$ = −1.12, 95% CI = −1.89 to −0.35, $p = 0.009$; see Supplementary Table 3 for full model output). *Post hoc* comparisons revealed that this interaction was driven by two effects. First, there was greater overall acceptance of effort when facing reward-versus-effort compared to threat-versus-effort trade-offs ($p < 0.001$ for both the $MAST_{PLC}$ and $MAST_{EXP}$ groups), suggesting different indifference points between experiments likely attributable to task design. Secondly, and most importantly, we observed that acute stress increased acceptance of effort for threat-versus-effort trade-offs ($p = 0.008$, $\chi^2 = 7.00$), but found no evidence

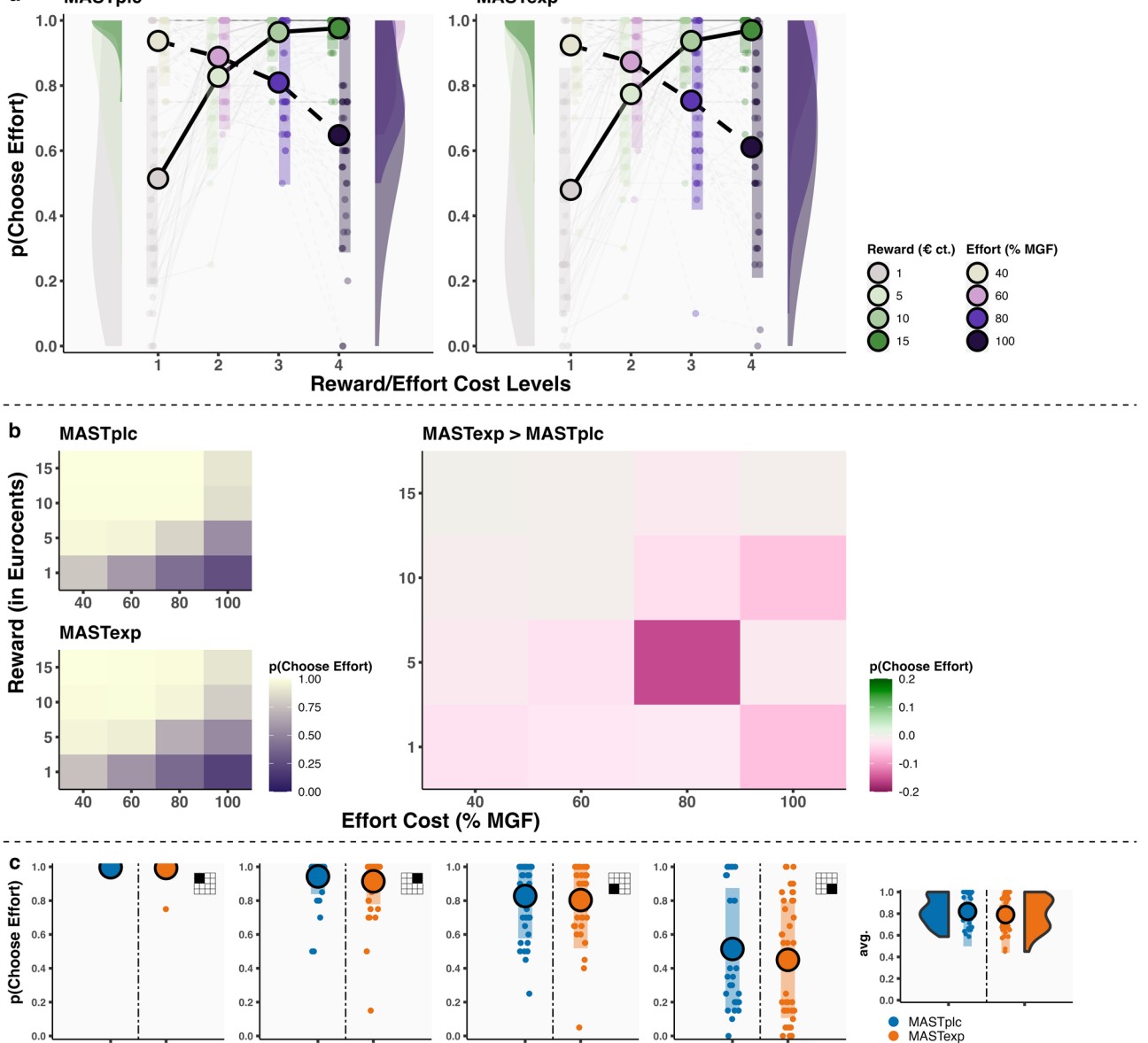

**Fig. 6 | High striker choice data. a** In Experiment 2, effort cost and reward value levels (*x*-axis) were negatively/positively associated with a greater likelihood of accepting effort (*y*-axis). **b** Heatmaps of average effort acceptance for the 16 unique offers that can be created by combining the 4 effort cost (*x*-axis) and 4 reward (*y*-axis) levels. Shown for $MAST_{PLC}$ and $MAST_{EXP}$ participants separately (left, in white-to-purple) and as a group difference score (right, in green-to-pink). **c** Effort acceptance (*y*-axis) for low/high effort/reward offers (see each plot's inset). Note: all plots are based on raw participant data. Error bars represent SD. $n = 84$ participants.

that acute stress increased acceptance of effort for reward-versus-effort trade-offs ($p = 0.19$, $\chi^2 = 1.68$). Taken together, these results do not suggest that acute stress promotes a more general willingness to exert effort. Instead, in line with its defensive function, we observed a selective increase in the willingness to exert effort in the service of avoiding threats.

Accounting for calibrated force levels, excluding high reward/low effort trials (which were accepted by almost all Experiment 2 participants), and excluding inflexible responders (Experiment 2: $n = 7$ MAST$_{PLC}$, $n = 3$ MAST$_{EXP}$) did not change the above-mentioned results (analyses available in accompanying code).

## Discussion

How does the acute stress response facilitate decisions and actions that prioritize survival? Using a novel effort-threat trade-off task, we aimed to shed light on the motivational processes that may underlie such behavior. Specifically, we observed that participants exposed to a stress-induction manipulation exhibited an increased willingness to exert physical effort in exchange for avoiding a threat-of-shock. Moreover, following the decision to exert effort, these participants exerted greater force intensity while trying to neutralize the threat-of-shock. A second experiment, and direct comparisons between experiments, provided no evidence for a stress-induced change in the willingness to exert effort for monetary rewards, suggesting that acute stress does not promote a more general willingness to exert effort. In the presence of threat, the acute stress response facilitates a bodily and psychological state in which available resources are strategically allocated to enhance survival. This is assumed to involve prioritization of threat processing, salience, and vigilance over reliance on computationally intensive executive control strategies[2,5,11]. Results from Experiment 1 and 2 complement this view by showing that the acute stress response can facilitate a motivational state that is characterized by an increased willingness to avoid aversive outcomes (i.e., aversive motivation).

At the computational level, this motivational state may involve alterations in the subjective value computations that govern the trade-off between minimizing effort (e.g., energy expenditure) and maximizing safety—two behavioral controllers that influence many of our everyday decisions. Specifically, computational modeling analyses suggested that the subjective value computations of acutely stressed participants were biased towards exerting effort for threats that were unlikely to result in an electric shock. These changes in integrative effort-threat subjective value computations may be attributable to multiple different underlying mechanisms. A greater willingness to exert effort to avoid unlikely threats in acutely stressed participants may have been facilitated by less steep discounting of higher (versus lower) levels of effort. Although such a hyperbolic discounting function should exert general effects on choice behavior (for which we observed subtle evidence; Fig. 3b), in the context of our paradigm, it should especially affect choice behavior on offers with the most negative objective value (e.g., low threat/high effort). This is because most participants are, naturally, unlikely to exert effort on these uneconomical offers, in which—assuming linear levels—effort cost heavily outweighs threat-of-shock. Furthermore, previous work has reported increased fear generalization[53], salience of threats[54], and a decrease in the ability to adaptively regulate fear[7,55] under stress. Arguably, such mechanisms will also increase the aversive value that we assign to threats, as well as the subsequent mobilization of force with which we try to neutralize them.

Another possibility is that our acute stress challenge may have impacted decision-making in the context of risk, uncertainty and/or potential punishments. Previous work has investigated how acute stress affects risk-taking behavior, loss aversion, and ambiguity aversion[56–61]. Although this work has revealed generally mixed effects that are dependent on individual characteristics and stressor type[58,61–63], a number of studies in this domain have demonstrated that acute stress can increase risk-taking behavior[64–67]. Moreover, combined glucocorticoid and noradrenergic activation, as is common during acute stress, can reduce aversion to losses[60]. Given that acutely stressed participants were

more, and not less, likely to exert effort to neutralize a probability of shock, and in light of the above-mentioned results, stress-induced increases in risk taking and/or reductions in loss aversion cannot fully account for Experiment 1 results.

Regardless of the exact mechanisms involved, observations of greater aversive motivation under stress advance our understanding of defensive behavior. Humans and other animals can choose from a range of defensive behaviors depending on the distance between threat and agent (i.e., defensive distance) and the perceived intensity of danger[6,22,68]. Defensive approach refers to anxiety-mediated behaviors directed at cautiously approaching threats (e.g., as in approach-avoidance dilemmas). Defensive avoidance, on the other hand, encompasses strategies that principally serve to escape threats, which has been thought to be fear-mediated and that are often, but not always, active in nature[22]. Our results speak to a kind of survival motivation that benefits defensive avoidance specifically; altered effort-threat subjective value computations may increase the willingness to carry out vigorous defensive actions that prioritize safety above all else. Although such behavior bears superficial similarity to more erratic, or uncoordinated, "flight" responses, it is worth emphasizing that vigorous avoidance actions can also be deliberative and/or planned[69].

Results from Experiment 2, as well as a limited amount of available previous work, do not support the idea that the acute stress response promotes a universal increase in the willingness to exert effort[12–14]. For example, stressed participants have been reported to exhibit a reduced willingness to exert cognitive effort, such as engaging in a challenging demand selection task[13]. On the other hand, we have previously observed a reduced emphasis on learning to minimize physical effort under stress (thus facilitating effort expenditure)[12], while Forbes et al.[14] reported a reduction in the willingness to exert physical effort only for the financial benefit of others. In contrast, the study by Forbes et al.[14] found no evidence for a stress-induced change in the willingness to exert physical effort for personal rewards, aligning with Experiment 2 results. Collectively, these emerging results reveal how stress may induce situationally-specific effects on motivation, which is relevant for psychopathology related to anxiety and/or avoidance, reward-seeking behavior, and social interactions. Moreover, this work offers important clues on central and peripheral stress-associated mechanisms that uniquely affect the willingness to exert physical versus cognitive effort (e.g., reduced availability of resources for complex cognitive functions[11]). At the same time, the importance of between-study differences must also be acknowledged, given that the exact subjective value computations employed strongly depend on the type of effort[18,19], the other option on offer[39], outcome valence[70,71], and interindividual differences (e.g., in aversion to effort, reward/punishment sensitivity, and fear of aversive outcomes[39,58]). On this note, it is worth mentioning that Experiment 1 and 2 differed not only in reinforcer type (i.e., shock versus reward), but also in the way that these variables were manipulated. Specifically, Experiment 1 used varying probabilities of experiencing a shock with pre-calibrated intensity, whereas in Experiment 2 guaranteed rewards varied in magnitude. While our findings provide evidence for a stress-induced increase in the willingness to exert effort in the service of avoiding threats, there is the possibility that Experiment 1–2 differences may, at least partially, stem from divergent effects of acute stress on decisions involving threat probability versus reward magnitude. Indeed, some evidence points toward reduced reward sensitivity under stress as measured by reward learning and motivation paradigms[72–74], again underscoring the possibility of reinforcer and/or situationally-specific effects of acute stress on the willingness to exert effort.

To conclude, the interplay between stress, effort valuation, and avoidance is crucial to understanding how we adaptively respond to threats. Here, we provide evidence for one such adaptive mechanism, in the form of prioritization of safety over minimization of effort expenditure. These results shed light on how stress can shape the motivation to engage in avoidance behavior.

**Article**

## Limitations

While the current study yields insights into the motivational processes impacted by acute stress, it is important to acknowledge outstanding questions and limitations that will need to be addressed in future research. First, our study design did not allow us to precisely differentiate between stress phases. The initial "stress reactivity" phase coincides with the almost-immediate and brief release of catecholamines and gradual rise of cortisol, while "stress recovery" is thought to occur during the gradual decrease of cortisol levels back to pre-stressor baseline[11,75]. Because all participants started the effort-based decision-making task immediately post-MAST, the great majority of task trials were completed at an early stage of the stress response, during which we also observed heightened (adrenaline-mediated) arousal and rising cortisol levels. However, a study design that allows for a more precise distinction between stress phases, or one that can disentangle the role of catecholamines versus glucocorticoids, will provide a deeper understanding of stress phase-dependent changes in the willingness to exert effort under threat. Secondly, the smaller offer sampling space—necessary to complete the experiment within the early stage of an acute stress response—resulted in ceiling effects for a subset of offers in Experiment 2. Although Experiment 2 results were robust to various *post hoc* checks, more comprehensive calibration of indifference points and/or use of a larger sampling spaces are recommended for future studies. Third, although experiments and conditions were demographically strictly balanced, the entire study population exhibited a sex imbalance and no constraints on contraceptive status were used. In light of sex and contraceptive-associated effects on cortisol stress reactivity[76,77], these imbalances could account for some of the observed results. Finally, while our study was adequately powered to detect more general effects of acute stress on the willingness to exert effort (see "Participants" section), it was not optimally powered to detect complex three-way interactions. Observed condition differences in highly selective parts of the paradigm sampling space should therefore be interpreted with caution and replicated in future work.

## Data availability

All data presented in this manuscript are accessible via the following link: https://osf.io/7pukd/.

## Code availability

All analyses presented in this manuscript are accessible via the following link: https://osf.io/7pukd/

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

## Acknowledgements
We thank Drs. Elen Romero Cifuentes, Drs. Anna-Maria Sakellariou, Drs. Dimitra Barmpari, Drs. Maria Odenheimer Daher, and Jenna Hendey for involvement in participant recruitment. We are indebted to Dr. Maud Daemen, Drs. Nele Volbragt, and Truda Driessen for administrative support. The authors received no specific funding for this work.

## Author contributions
Kristína Pavlíčková: development of methodology, programming, analysis, data curation, writing. Judith Gärtner: development of methodology, programming, data collection, writing. Stella D. Voulgaropoulou: development of methodology, conceptualization, programming, visualization, writing, project administration. Deniz Fraemke: development of methodology, conceptualization, programming, validation, writing. Eli Adams: development of methodology, conceptualization, programming, data collection, writing. Conny W.E.M. Quaedflieg: development of methodology, writing, project administration. Wolfgang Viechtbauer: development of methodology, analysis, writing, supervision. Dennis Hernaus: development of methodology, conceptualization, analysis, data curation, writing, visualization, supervision, project administration, funding acquisition.

## Competing interests
The authors declare no competing interests.
