## [Peer Review File · Communications Psychology]

14th Feb 24

Dear Dr Hernaus,

First of all, Thank you for your patience during this exceptionally long peer-review process. Your manuscript titled "The Drive to Survive: Acute Stress Promotes Aversive Motivation" has now been seen by 3 reviewers, and I include their comments at the end of this message. They find your work of interest but raised some important points. We are interested in the possibility of publishing your study in *Communications Psychology*, but would like to consider your responses to these concerns and assess a revised manuscript before we make a final decision on publication.

We therefore invite you to revise and resubmit your manuscript, along with a point-by-point response to the reviewers. Please highlight all changes in the manuscript text file.

Editorially, we consider it important that the following aspects are carefully addressed:

First, the three reviewers raise concerns about statistical power, please add a sensitivity analysis on the main effects of interest for the conclusion.

Second, the reviewers consistently ask for more details about experiment 2, we want to see experiment 2 fully reported in the main manuscript with its methodological and statistical details.

Third, the reviewers raise serious doubts about the comparability of experiment 1 and experiment 2. In particular, Reviewer 3 elaborates several aspects that limit the comparability such as the nature of the reinforcers (primary vs secondary) or the manipulation of reinforcer intensity vs reinforcer probability. Editorially, we agree that the two studies do not appear to be directly comparable, and we would like the main conclusions of the manuscript to focus on the significant results of experiment 1. Please also add Bayes Factors or equivalence tests to the null results in experiment 2.

Finally, Reviewer 2 raises important concerns about the modelling of the error term in the multi-level regression, please carefully address this important point.

I am attaching an Editorial Requests Table that details critical reporting requirements for the revised manuscript. Please attend to each item and ensure your manuscript is fully compliant. We are requesting that your manuscript aligns with these requirements as this facilitates the evaluation of your manuscript, reducing delays in re-review and potential future acceptance. If your revised manuscript is not aligned with these requests on major issues, such as those concerning statistics, it may be returned to you for

further revisions without re-review. Additional information can be found in our style and formatting guide Communications Psychology formatting guide.

Please use the following link to submit your

- revised manuscript,
- point-by-point response to the referees' comments,
- cover letter (as a separate document),
- the Editorial Policy Checklist (see below),
- the Reporting Summary (see below), and
- the completed Editorial Request Table (attached):

[link redacted]

Best regards,

Eva R. Pool

Eva R. Pool, PhD

Editorial Board Member

Communications Psychology

orcid.org/0000-0001-5929-1007

REVIEWER EXPERTISE:

Reviewer #1 Stress and Cognition

Reviewer #2 Stress, Motivation and decision making

Reviewer #3 Computational modeling of decision making

REVIEWER REPORTS:

Reviewer #1 (Remarks to the Author):

Congratulations on this highly relevant work! The results are very interesting and certainly a very important contribution to this field. The findings on stress on effort exertion have been mixed, pointing to the importance of certain boundary conditions/task-specific differences, like physical versus cognitive effort exertion. The authors show that it matters whether the goal is to gain reward or to avoid threat. Under stress, the willingness to exert effort is increased when the goal is to avoid threat, whereas this is not the case when the goal is to gain reward. The use of a computational approach points to a prioritization of safety over energy conservation under stress, which nicely adds mechanistic insights. The concept of effort and the effects of stress on effort exertion have lately gained interest in the scientific community and I think this publication will add relevant information to the ongoing discussions. The study design is very well thought out (even though the authors acknowledge some limitations), including the use of dextrose and an independent follow-up experiment. The statistical analyses and performed checks are sound, the results are convincing, and they will be of interest to other researchers in the field. Sufficient details have been provided in order for other researchers to reproduce the work. All data and analyses have been made openly available. Therefore, I recommend to accept the manuscript, after the following editorial revisions have been provided.

1. Age 16-35: did inclusion of under-age participant involve a different consent procedure?

2. Lines 72-73: 80 participants were randomized to a validated stress induction procedure that robustly upregulates activation of major stress axes  mention here that 40 were subjected to stress and 40 were subjected to control condition
3. Figure 2 should be bigger than currently
4. Figures: indicate what the error bars show
5. Was the N the same for all statistical analyses or were any additional participants excluded from only certain analyses?
6. I am not sure the way you report statistics always conforms with the guidelines, for the frequentist inferential statistics I believe the confidence intervals are missing (% Confidence Intervals = values). Please adjust where appropriate.
7. Please follow the author guidelines and add this information: If there was no preregistration of any study, this must be declared in the Methods section for transparency.
8. Can you refer in the methods to the supplements for details about Experiment 2? Where the details can be found is mentioned quite late in the manuscript and might cause confusion.
9. Figure 3: difficult to see what text belongs to which axis in what part of that figure. Add in legend information: e.g. y-axis of heat maps = threat-of-shock (%), x-axis of heatmaps = effort cost (% MGF). And/or try to make this clearer in the figure, for example by increasing the distance between the sub-figures or making the distinction between sub-figures bigger by using background colors etc..
10. "we investigated associations between choice behavior and MAST-induced (negative) affect, autonomic (systolic blood pressure), and glucocorticoid (sCORT) responses by adding these variables as predictors to the model." If you added them as pre vs. post MAST values here can you add this information?

Reviewer #2 (Remarks to the Author):

The current paper is concerned with the motivational effects of acute stress on decision-making. As such, it is a timely and important contribution to the literature. The authors used a grip force task to measure effort assertion (or energy expenditure) to test this. Here, Pavlíčková et al. could show that effort assertion is highly selective and depends on the importance one ascribes to the situation.

First, I compliment the authors; this manuscript was a pleasure to read. Overall, the methodology appears to be expertly done, including stress and computational modelling—and I do not see any significant problems with the approach or the interpretation of data. This is an important achievement as this research is inherently complicated.

However, second, there are some comments that I want to bring attention to that should be considered before I can recommend this manuscript for publication in *Communications Psychology*. In particular, I wonder if these studies are adequately powered. Two rather low-powered studies are compared—which is less than ideal for drawing firm conclusions. Even with low power, these studies are still publishable, as they provide novel insights into an important (and still emerging) field in stress research. As such, they will surely make an impact. While this and other considerations are important to address, I am confident that the authors can address the outstanding questions and issues sufficiently.

Major comments

At first glance, the combined sample sizes ($n \sim 160$) might seem sufficiently large. However, it should be noted that the individual samples are rather small—particularly for testing triple interactions (even with this many trials). I have doubts that this study was adequately powered, which gives me lower confidence in the results, especially since there are two low-powered studies, and drawing firm conclusions on the comparison between these studies seems a bit odd. It is not that I doubt the data—or the argument the authors make—but I would want this addressed somewhere (also see the following comments).

Study 1 used the “Effort-Threat “Thunderstorm” Task”, but what task was used for study 2? This needs to be explained in more detail, especially if one wants to determine differences and similarities between the studies.

The authors chose to test a sample including both men and women. Including both men and women is commendable. However, there are some problems with this approach. First, the samples are very imbalanced (with more women than men), making the probing of gender/sex-specific effects rather difficult. Furthermore, the effects of women are conflated with a mix of OC and cycling women. I bring this up because the human stress response differs between men and women, and within the menstrual cycle for women, and for regularly cycling women compared to OCs—as the authors are undoubtedly familiar with (as these have been considerations in the past). Given the small sample, this is a limitation of the current approach and needs to be addressed in the limitations of the current manuscript.

The statistical models could be clearer to me, and I need additional information or clarification. For example, it is unclear how the regressions were defined; what were the DVs and interactions? How many trials were included (all trials of the task P9 L276, or just the aggregated behaviour from the trial-blocks, P9 L283)? Furthermore, the error structure for the mixed-effects model is unclear. Was the random-error structure participant-ID as a random intercept only? This would seem a bit too simplistic to model this kind of nested data (cf. Barr 2013; J Mem Lang). However, the authors also talk about “all three predictors were added as random and fixed effects” (P9 L286). I assume as random slopes? And when was this the case? (For all models?). Was this additive or multiplicative (as for the triple interaction)?

A table with the regressions would be helpful (even in the Supplementals).

While I find the arguments convincing, I would be careful when comparing these findings with too much confidence. While the two study designs are very similar, questions remain on how comparable they are. For example, I am unsure how comparable the two tasks are: Is it just about monetary rewards? (Some additional info on the second task might be helpful in this regard). Are the two samples comparable? (Maybe?). Etc. I would address this in the limitation section of the paper.

Minor comments

How did the authors test eligibility based on the DSM-V? Was this a self-reported measure?

The choice of administering dextrose before stress is an interesting one. This approach introduces more confounds than a spontaneous, unaltered HPA axis response. Yes, a manipulated HPA axis response might look nicer on paper, but there is insufficient evidence to attribute this spike in cortisol solely to a higher metabolic potential. Other factors, like the duration of fasting (only 2 hrs, a rather short fastening period is controlled for here, but the actual time might differ between subjects), gender/sex-specific effects (as mentioned by Zänker et al.), dosage effects (the cited studies vary greatly, and there is a range in body compositions in this study as well BMI >18 & BMI <30) All of this is to say, this approach makes studies less comparable and glucocorticoid-specific effects less confident. I do not know if this approach actually standardizes metabolic output potential.

That said, I appreciate the disclosure, and this transparency is sufficient to address methodological differences in stress induction, particularly in the glucocorticoid stress response between studies. I do not need this to be addressed any further.

On page 18 L580, before discussing Forbes et al, there seems to be a missing citation (we found that ...)

Reviewer #3 (Remarks to the Author):

The current study examined the effects of acute stress on physical effort exertion in the context of threat avoidance and, in a second experiment, reward seeking. The authors report that stress increased participants' willingness to expend effort to avoid receiving electrical shocks but did not change behavior meaningfully when effort exertion would confer additional rewards.

The manuscript is well-written and tackles an important question that perfectly fits the scope of the journal and should be of great interest to its readership. I enjoyed reading the manuscript and I do not have any serious issues with the design that would require substantial changes before publication. Nonetheless, I have a number of questions/concerns that I think could use clarification. I will list my comments separately for each section (order does not relate to severity of the comment).

Introduction:

Overall, I think the introduction is very clear. I only have two minor comments.

- The authors, throughout the manuscript, frequently use the term 'energy', often interchangeably with effort or resources. I do understand the (very intuitive) appeal but it is unclear what energy specifically refers to with respect to effort-based choices. Admittedly, this problem is bigger when talking about mental effort but even in the physical domain, it is still debatable what these energetic resources or costs really represent (e.g., ATP in muscle cells? Glucose? The feeling of fatigue?). Thus, I would personally refrain from using 'energy' and replace it either with more specific terms or other general terms that do not necessarily imply a metabolic cost. That being said, this is not a hill I am willing to die on, so if the authors think 'energy' is the best word to use, then they are of course free to do it.

- The authors end the introduction with a summary of their results. This is unusual (although it looks like it has become more common recently) and maybe something the journal encourages (?), but I do not think it's necessary to foreshadow the rest of the paper and could be taken out. But again, this is a decision I will ultimately leave to the authors and the editor.

Methods:

- As far as I could see, there was no specific sample size justification provided by the authors. How did they arrive at $n = 80$ (and $n = 84$) as their target sample size?

- On page 6, the authors use the terms SCORT and sAA without properly introducing these abbreviations before.

- In line 148, the authors state that they measured heart rate at -10 and 0 min (in relation to the MAST). Were these the only times heart rate was measured? If so, why was it not measured after stress? (Although the timing might be in relation to stress offset instead of onset? see my comments below).

- The authors used a novel task to test their primary hypothesis. While I appreciate the overall task design, one of my bigger concerns about this study relates to the interpretation of the results from this task as well as how they relate to those of experiment 2. Specifically, in the 'thunderstorm' task, participants are asked to choose between exerting effort or receiving an electrical shock with a certain probability. Given that the shock, in contrast to the physical demand, has a constant intensity and only varies in its probability, could it be possible that stress did not (only) change participants' evaluation of effort but how they process these probabilities? I know the authors discuss this to a degree, namely in terms of a lack of evidence for changes in (monetary) risk taking under stress. However, some studies do suggest that this can be the case (e.g., Starcke et al., 2008; Nowacki et al., 2019; Pighin et al., 2020), so I do not think it can be fully excluded as a potential explanation for the results. I was wondering if the authors had specific reasons to use shock probability instead of intensity as an independent variable here.

- Moreover, the authors suggest that the results obtained in experiment 2 indicate that stress selectively increases motivation to exert effort in the aversive context of expecting shocks but not in a context in which participants exert effort to obtain rewards. However, I am not sure that the results can be compared that easily. In contrast to the shocks in experiment 1, rewards in experiment 2 were varied by amount (or reward intensity) but not by the probability of receiving them. In other words, the effort-associated rewards and punishments were manipulated in distinct ways across both experiments, namely outcome magnitude vs. outcome probability, respectively. Could it be possible that stress might have affected effort-based decisions about reward if the authors had used a fixed amount (e.g., 15 cents) and varied the probability of receiving it? To be clear, I do not necessarily doubt the validity of each experiment's finding but think that the conclusion that stress only specifically affect motivation to exert effort in one but not the other context might not be fully warranted due to the differences in task design outlined here.

- In addition to the point above, the strength of the conclusion that stress affects aversive but not appetitive motivation suffers a bit from the fact that the two experiments used different participants. It is possible that choice behavior of individuals participating in experiment 1 was simply more sensitive to stress than the choice behavior of participants in experiment 2.

- From the authors' descriptions, I did not fully understand at which timepoint, relative to the MAST, the 'thunderstorm' task was administered. Given the temporal profile of the physiological stress response(s), this information is crucial to discern how exactly stress may have affected behavior. It might be helpful to highlight (maybe in the lower panel in figure 2) when exactly the task took place. Similarly, it should be clarified when the MAST started and ended. I initially assumed 't+00' would indicate the time of stress onset but given the apparent difference in cortisol at that time and that heart rate measures are labeled as pre/post in the figure but as 't-10' and 't+00' in the text, it is possible that 't+00' actually refers to stress offset. Is that correct?

- To further elaborate this point, the authors state that the task is supposed to take place in the 'fight-or-flight' stage of the stress response but do not specify what they mean by that. Usually, this term is primarily associated with the initial SAM-dependent response resulting in increased secretion of catecholamines but here it might be meant to extend to encompass HPA axis activation as well? The distinction is important because SAM activation, as the term 'fight-or-flight' indicates, is thought to facilitate the immediate response to a stressor and normalizes almost instantly after the organism has escaped (or defeated) the threat (which may explain the lack of sAA differences pre/post MAST here). In contrast, glucocorticoid actions are more thought to handle the aftermath of stressful experiences. Although there's an overlap between both systems during the down ramping of the SAM and the up ramping of the HPA axis, this time-window is very brief. The task, on the other hand, took 27 minutes in total and was presumably completed outside the specific context of the MAST (i.e., after a short delay to collect physiological measures and maybe even in a different room?). How did the authors make sure the effects can mostly be related to the early 'fight-or-flight' stage of the stress response? Either way, I think those timing issues and their potential consequences for the mechanisms underlying the observed stress effect could be discussed in a bit more detail.

Results:

- Line 396: the authors refer to figure 1A but I think this should be 3A.

- Why did the authors choose to group the different effort/threat-of-shock combinations into a 2x2 comparison matrix for their post-hoc analyses instead of looking at all 4x4 comparisons (figure 3B)? Was it to increase the amount of data that goes into each comparison or maybe to reduce the amount of post-hoc tests to correct for in the analysis? I do not have a problem with any of these approaches, but it should be stated for transparency and reproducibility purposes.

- Looking at Figure 3B, the largest differences (and probably the driving factor of the significant result) are found in the 80% effort/20% threat-of-shock bin and the 80% effort/30% threat-of-shock bin, where the stress group displayed much higher motivation to exert the effort. More generally, the differences between the groups seems to be most pronounced at intermediate effort levels (60 and 80%) instead of the high effort/low probability end of the scale as discussed by the authors (e.g., the 100% effort/20 and 100% effort/30% probability bins are almost identical between groups). To me, the stronger sensitivity to stress particularly in these in-between demand levels could suggest that participants operated close to their individual indifference points in these trials. As such, it could be argued that stress specifically tips the scale in situations in which participants have a hard time deciding between options. I would love to read the authors' thoughts on this slightly different interpretation of their findings.

- For experiment 2, the authors report main effects and (non-significant) condition-related interaction effects, but no other tested effects. I understand that the main text would become difficult to read if effects from all predictors were included but it would be great to add a (supplemental?) table that lists all main and interaction effects from the regression models used in the study (including experiment 1).

- With respect to the statistical comparison of the stress effects between studies, the authors state that stress only increased motivation in the threat but not the reward context. At the same time, they acknowledge that under reward prospect, individuals were more likely to exert effort in general. Is it possible that the lack of increase in effort exertion under stress in experiment 2 simply reflects ceiling effects?

- Although not significant, it struck me as interesting that descriptively, stress seemed to reduce motivation to exert effort in experiment 2, which would fit other work showing increased effort aversion under stress. In line with experiment 1, the strongest negative effect was again at a more intermediate effort level (80% effort/5 cents reward). I would love to read the authors' thoughts on this (again, being aware that this does not come out as statistically significant).

Discussion:

- Lines 525/526: The authors state that 'Moreover, following the decision to exert effort, the same participants exerted greater force intensity while trying to neutralize the threat-of-shock.' From the results, this looked like a simple group difference comparison. Was it indeed the very same participants (i.e., was there a correlational relationship where participants with stronger effort aversion were the ones displaying the highest force intensity)?

- Line 529: 'promotes' should be 'promote'

EDITORIAL POLICIES

We ask that you ensure your manuscript complies with our editorial policies and reporting requirements.

To that end, we require revised manuscripts to be accompanied by two completed items: a reporting summary that collects information on study design and procedure, and an editorial policy checklist that verifies compliance with all required editorial policies.

Nature Research Reporting Summary

Editorial Policy Checklist

All points on the policy checklist must be addressed. Your revised manuscript can only be sent back to the referees if these checklists are completed and uploaded with the revision.

Notes: If you have submitted a Stage 1 Registered Report, Review, Primer, Comment, or Perspective you do not need to submit these forms. If you have already submitted these forms, you may disregard this request.

Authors reply to Reviewer Comments on COMMSPSYCHOL-23-0396

We thank the Reviewers for their comments. Below we provide a point-by-point reply to all comments raised. All changes in the manuscript are highlighted via track changes. Where applicable, we include the updated text (in red) into the rebuttal letter. While addressing the Reviewer comments, some edits were made to analysis scripts. Updated analysis scripts have been uploaded to a new project folder on the OSF (<https://osf.io/7pukd/>, ScriptsReadme >> v1). An update log is included so that readers can see which script edits were made.

Reviewer #1 (Remarks to the Author):

Congratulations on this highly relevant work! The results are very interesting and certainly a very important contribution to this field. The findings on stress on effort exertion have been mixed, pointing to the importance of certain boundary conditions/task-specific differences, like physical versus cognitive effort exertion. The authors show that it matters whether the goal is to gain reward or to avoid threat. Under stress, the willingness to exert effort is increased when the goal is to avoid threat, whereas this is not the case when the goal is to gain reward. The use of a computational approach points to a prioritization of safety over energy conservation under stress, which nicely adds mechanistic insights.

The concept of effort and the effects of stress on effort exertion have lately gained interest in the scientific community and I think this publication will add relevant information to the ongoing discussions. The study design is very well thought out (even though the authors acknowledge some limitations), including the use of dextrose and an independent follow-up experiment. The statistical analyses and performed checks are sound, the results are convincing, and they will be of interest to other researchers in the field. Sufficient details have been provided in order for other researchers to reproduce the work. All data and analyses have been made openly available. Therefore, I recommend to accept the manuscript, after the following editorial revisions have been provided.

R1, Comment 1. Age 16-35: did inclusion of under-age participant involve a different consent procedure?

- **Reply:** The consent procedure was identical for all participants. In The Netherlands, competent participants age 16 and older are legally allowed to provide informed consent.

R1, Comment 2. Lines 72-73: 80 participants were randomized to a validated stress induction procedure that robustly upregulates activation of major stress axes  mention here that 40 were subjected to stress and 40 were subjected to control condition

- **Reply:** We now clarify that each condition contained 40 participants. Text now reads:

“To test this hypothesis, 80 participants were randomized to a validated stress-induction procedure that robustly upregulates activation of major stress axes²⁷ (n=40), or no-stress control condition (n=40), after which [...]”

R1, Comment 3. Figure 2 should be bigger than currently

- **Reply:** We have included larger versions of the MAST panel figures in the new version of this manuscript.

R1, Comment 4. Figures: indicate what the error bars show

- **Reply:** All error bars in the manuscripts represent standard deviation. This information is now included in all figure legends.

R1, Comment 5. Was the N the same for all statistical analyses or were any additional participants excluded from only certain analyses?

- **Reply:** All analyses involving demographics, stress measurements, task choice data, and task grip force data were conducted on n=80 (Experiment 1) and n=84 (Experiment 2) participants. Only for the robustness analysis involving exclusion of Experiment 2 “inflexible responders” did the sample naturally deviate, but here we explicitly mention in text how many were left out (i.e., n=7 MAST_{PLC}).

n=3 MAST_{EXP}). The final sentence of the “Statistical Analyses” section now clarifies the n used in all analyses:

“Unless explicitly stated otherwise, all reported results are based on the entire sample of n=80 (for Experiment 1 analyses) and n=84 (for Experiment 2 analyses).”

R1, Comment 6. I am not sure the way you report statistics always conforms with the guidelines, for the frequentist inferential statistics I believe the confidence intervals are missing (% Confidence Intervals = values). Please adjust where appropriate.

- **Reply:** We thank the Reviewer for pointing this out. We checked the Editorial Request Table and indeed noticed that the reporting of certain inferential statistics was not in line with journal guidelines. The following edits have been made:
 - Tables of demographics/self-report questionnaire scores: added 95% confidence intervals for group differences in proportions (for Chi Square tests) and mean values (for t-tests).
 - MAST stress induction: added 95% confidence intervals for all baseline statistical tests (i.e., group differences pre-MAST) and *post hoc* tests (timepoint differences stratified by group, group differences stratified by timepoint).
 - Diagnostic checks: added 95% confidence intervals when reporting relationships between/group differences in calibration values and choice behaviour.
 - GLM results: clarified that all reported values in brackets represent 95% CIs.

R1, Comment 7. Please follow the author guidelines and add this information: If there was no preregistration of any study, this must be declared in the Methods section for transparency.

- **Reply:** We have added the following statement to the final paragraph of the “Statistical Analyses” section:

“The statistical analyses and models reported above were not pre-registered.”

R1, Comment 8. Can you refer in the methods to the supplements for details about Experiment 2? Where the details can be found is mentioned quite late in the manuscript and might cause confusion.

- **Reply:** We agree with the Reviewer’s suggestion and have therefore reported all Experiment 2 information in the main text. This includes:
 - Sample inclusion/exclusion criteria
 - Reward-effort paradigm details
 - Statistical models
 - Results related to stress-induction and/or task behaviour
 - Relevant figures (e.g., task figure, stress-induction plot)

R1, Comment 9. Figure 3: difficult to see what text belongs to which axis in what part of that figure. Add in legend information: e.g. y-axis of heat maps = threat-of-shock (%), x-axis of heatmaps = effort cost (% MGF). And/or try to make this clearer in the figure, for example by increasing the distance between the sub-figures or making the distinction between sub-figures bigger by using background colors etc..

- **Reply:** We have now updated this figure and the same figure for Experiment 2. In short, we have added dashed lines between each panel and increased the axis font to emphasize the different panels and units in the figures. The legend text has also been thoroughly revised.

R1, Comment 10. “we investigated associations between choice behavior and MAST-induced (negative) affect, autonomic (systolic blood pressure), and glucocorticoid (sCORT) responses by adding these variables as predictors to the model.” If you added them as pre vs. post MAST values here can you add this information?

- **Reply:** The Reviewer’s assumption is correct. We indeed included stress parameter difference scores (post minus pre) and we now report this information under “Statistical Analyses”:

“When investigating associations between choice and MAST-induced (negative) affect, autonomic (systolic blood pressure), and glucocorticoid (sCORT) changes (i.e., post minus pre), the same multilevel model structure was used [...]”

Reviewer #2 (Remarks to the Author):

The current paper is concerned with the motivational effects of acute stress on decision-making. As such, it is a timely and important contribution to the literature. The authors used a grip force task to measure effort assertion (or energy expenditure) to test this. Here, Pavlíčková et al. could show that effort assertion is highly selective and depends on the importance one ascribes to the situation. First, I compliment the authors; this manuscript was a pleasure to read. Overall, the methodology appears to be expertly done, including stress and computational modelling—and I do not see any significant problems with the approach or the interpretation of data. This is an important achievement as this research is inherently complicated.

However, second, there are some comments that I want to bring attention to that should be considered before I can recommend this manuscript for publication in *Communications Psychology*. In particular, I wonder if these studies are adequately powered. Two rather low-powered studies are compared—which is less than ideal for drawing firm conclusions. Even with low power, these studies are still publishable, as they provide novel insights into an important (and still emerging) field in stress research. As such, they will surely make an impact. While this and other considerations are important to address, I am confident that the authors can address the outstanding questions and issues sufficiently.

Major comments

R2, Comment 1. At first glance, the combined sample sizes (n~ 160) might seem sufficiently large. However, it should be noted that the individual samples are rather small—particularly for testing triple interactions (even with this many trials). I have doubts that this study was adequately powered, which gives me lower confidence in the results, especially since there are two low-powered studies, and drawing firm conclusions on the comparison between these studies seems a bit odd. It is not that I doubt the data—or the argument the authors make—but I would want this addressed somewhere (also see the following comments).

- **Reply:** The Reviewer is correct in mentioning the importance of adequate statistical power. In previous work, we and others have observed that acute stress manipulations result in small-to-medium effects on effort-based decision-making tasks (partial eta squared/ η^2 of ~0.06; see e.g., Voulgaropoulou *et al.* 2022 <https://doi.org/10.1016/j.psyneuen.2021.105646> or Bogdanov *et al.* 2021 <https://doi.org/10.1177/09567976211005465>). Based on these studies, and assuming a comparable effect size of $\eta^2=0.065$ (which translates to ~10% group difference in effort acceptance; SD of 12), a sample of 40 participants in each cell (i.e., 160 in total) provides acceptable power (0.88) to detect the critical Experiment*Condition interaction on choice (assuming $\alpha=0.05$).

At the same time, we note that the power to detect the critical Condition*Effort*Threat interaction on choice in Experiment 1 is lower (power=0.59). However, it's important to consider that we also find 2-way interactions in Experiment 1, and even (simple) main effects of condition in Experiment 1 (e.g., in direct experiment 1-2 comparisons). We therefore believe that the current study is adequately powered to detect more general effects of acute stress on choice behaviour within and between experiments, but admit that highly specific effects (i.e., in specific locations of the offer sampling space) should be interpreted with caution.

Based on the above considerations, we have updated the manuscript. First, we have now added the above-mentioned power calculations for the Experiment*Condition and Condition*Effort*Threat interactions on choice to the "Participants" section. Secondly, in the Limitations section, we have underlined that reported 3-way interactions should be interpreted with caution. We hope that these edits sufficiently address the Reviewer's concern.

In the "Participants" section we have added the following text:

"The study was designed to yield sufficient power to detect more general medium effects of acute stress on the willingness to exert effort within and between experiments, although the power to detect more complex higher-order interactions with a similar effect size was reduced (e.g., with total n=160: power=0.88 to detect an

Experiment×Condition interaction on choice behaviour with effect size $\eta^2=0.065$ at $\alpha=0.05$; power=0.59 to detect a Condition×Effort×Threat interaction on choice behaviour)."

In the "Limitations" section we have added the following text:

"Finally, while our study was adequately powered to detect more general effects of acute stress on the willingness to exert effort (see "Participants" section), it was not optimally powered to detect complex three-way interactions. Observed condition differences in highly selective parts of the paradigm sampling space should therefore be interpreted with caution and replicated in future work."

R2, Comment 2. Study 1 used the "Effort-Threat "Thunderstorm" Task", but what task was used for study 2? This needs to be explained in more detail, especially if one wants to determine differences and similarities between the studies.

- **Reply:** We fully agree with this suggestion and have therefore incorporated all Experiment 2 in the main text (also see **R1/Comment 8**).

R2, Comment 3. The authors chose to test a sample including both men and women. Including both men and women is commendable. However, there are some problems with this approach. First, the samples are very imbalanced (with more women than men), making the probing of gender/sex-specific effects rather difficult. Furthermore, the effects of women are conflated with a mix of OC and cycling women. I bring this up because the human stress response differs between men and women, and within the menstrual cycle for women, and for regularly cycling women compared to OCs—as the authors are undoubtedly familiar with (as these have been considerations in the past). Given the small sample, this is a limitation of the current approach and needs to be addressed in the limitations of the current manuscript.

- **Reply:** We agree that study samples should ideally be balanced in terms of e.g., sex and contraceptive status. Because experiment 1 and 2 samples are highly comparable in terms of these, and many other, variables, we believe that between-study or between-group sex and contraception imbalance cannot explain the reported results. Nevertheless, such imbalances may reduce the generalizability of the results, and we now acknowledge this in the Limitations sections:

"Third, although experiments and conditions were demographically strictly balanced, the entire study population exhibited a sex imbalance and no constraints on contraceptive status were used. In light of sex and contraceptive-associated effects on cortisol stress reactivity^{77,78}, these imbalances could account for some of the observed results."

R2, Comment 4. The statistical models could be clearer to me, and I need additional information or clarification. For example, it is unclear how the regressions were defined; what were the DVs and interactions?

R2, Comment 5. How many trials were included (all trials of the task P9 L276, or just the aggregated behaviour from the trial-blocks, P9 L283)?

R2, Comment 6. Furthermore, the error structure for the mixed-effects model is unclear. Was the random-error structure participant-ID as a random intercept only? This would seem a bit too simplistic to model this kind of nested data (cf. Barr 2013; J Mem Lang). However, the authors also talk about "all three predictors were added as random and fixed effects" (P9 L286). I assume as random slopes? And when was this the case? (For all models?). Was this additive or multiplicative (as for the triple interaction)? A table with the regressions would be helpful (even in the Supplementals).

- **Reply (to Comment 4, 5, and 6):** We thank the Reviewer for suggestions to improve the clarity of the "Statistical Analyses" section, and have adjusted this section accordingly.
 - First, for all analyses, we have added what the DV, IV, and (if applicable), interactions were.
 - Secondly, we now explicitly mention the data structure (e.g., trials nested within participants, number of levels for a given variable) and have rewritten any parts that may inadvertently give the impression that we used aggregated (e.g., block-level) data in GLMs.

- Third, we clarified the error structure and are more consistent when using random/fixed effects terminology. We agree with the Reviewer that only adding subject as random intercept is too simplistic. In short, for main effects analyses, threat-of-shock/effort/block are added as random intercept and slopes to the subject level. In interaction analyses, condition was also added as a random intercept and slope to the subject level to allow for intercept variance, in line with Bates 2010 (lme4: Mixed-effects modeling with R, Chapter 2, Mixed Models Section) and because it explained meaningful variance (as indicated by improved model fit).
- Fourth, and also based on **R3/Comment 14**, we now include regression formulas in a new supplemental table, next to these formulas being available in the accompanying analysis code.

We hope that these additions and text changes enhance the clarity of the analyses.

R2, Comment 7. While I find the arguments convincing, I would be careful when comparing these findings with too much confidence. While the two study designs are very similar, questions remain on how comparable they are. For example, I am unsure how comparable the two tasks are: Is it just about monetary rewards? (Some additional info on the second task might be helpful in this regard). Are the two samples comparable? (Maybe?). Etc. I would address this in the limitation section of the paper.

- **Reply:** Reviewer 2 is correct in suggesting that Experiment 1 and 2 are not perfectly comparable. One important difference is that Experiment 1 manipulated threat probability, while Experiment 2 manipulated reward magnitude. We have made several changes to emphasize these between-experiment differences:
 - First, as also suggested by the Editor and Reviewer 1 (see **R1/Comment 8**), we have now fully incorporated Experiment 2 into the main text, making it easier for readers to directly contrast the two studies and samples (which score highly comparably on a wide range of demographic variables).
 - Secondly, Figure 1 now shows a direct comparison of Experiment 1 (threat) and Experiment 2 (reward) paradigms.
 - Third, in the discussion, we now speculate how between-experiment paradigm differences might account for some of the observed results:

“At the same time, the importance of between-study differences must also be acknowledged, given that the exact subjective value computations employed strongly depend on the type of effort^{18,19}, the other option on offer⁴⁰, outcome valence^{63,64}, and interindividual differences (e.g., in aversion to effort, reward/punishment sensitivity, and fear of aversive outcomes^{40,57}). On this note, it is worth mentioning that Experiment 1 and 2 differed not only in reinforcer type (i.e., shock versus reward), but also in the way that these variables were manipulated. Specifically, Experiment 1 used varying probabilities of experiencing a shock with pre-calibrated intensity, whereas in Experiment 2 guaranteed rewards varied in magnitude. While our findings provide evidence for a stress-induced increase in the willingness to exert effort in the service of avoiding threats, there is the possibility that Experiment 1-2 differences may, at least partially, stem from divergent effects of acute stress on decisions involving threat probability versus reward magnitude. Indeed, some evidence points toward reduced reward sensitivity under stress as measured by reward learning and motivation paradigms⁷³⁻⁷⁵, again underscoring the possibility of reinforcer and/or situationally-specific effects of acute stress on the willingness to exert effort.”

Minor comments

R2, Comment 8. How did the authors test eligibility based on the DSM-V? Was this a self-reported measure?

- **Reply:** The updated text in the “Participants” section now clarifies which criteria were confirmed using self-report measures.

“Healthy human participants that decided to take part in Experiment 1 or 2 were screened for eligibility using self-report questions about age (16-35), absence of a DSM (5th ed.)²⁹ psychiatric disorder, neurological disorder, diabetes type 1 or 2, [...]”

R2, Comment 9. The choice of administering dextrose before stress is an interesting one. This approach introduces more confounds than a spontaneous, unaltered HPA axis response. Yes, a manipulated HPA axis response might look nicer on paper, but there is insufficient evidence to attribute this spike in cortisol solely to a higher metabolic potential. Other factors, like the duration of fasting (only 2 hrs, a rather short fastening period is controlled for here, but the actual time might differ between subjects), gender/sex-specific effects (as mentioned by Zänker et al.), dosage effects (the cited studies vary greatly, and there is a range in body compositions in this study as well BMI >18 & BMI <30) All of this is to say, this approach makes studies less comparable and glucocorticoid-specific effects less confident. I do not know if this approach actually standardizes metabolic output potential.

That said, I appreciate the disclosure, and this transparency is sufficient to address methodological differences in stress induction, particularly in the glucocorticoid stress response between studies. I do not need this to be addressed any further.

- **Reply:** We thank Reviewer 2 for sharing their input on this design choice. Our short answer is that we agree with the notion that the great variation in guidelines and procedures in stress-induction studies makes direct comparisons challenging.

R2, Comment 10. On page 18 L580, before discussing Forbes et al, there seems to be a missing citation (we found that ...)

- **Reply:** Thank you – adjusted.

Reviewer #3 (Remarks to the Author):

The current study examined the effects of acute stress on physical effort exertion in the context of threat avoidance and, in a second experiment, reward seeking. The authors report that stress increased participants' willingness to expend effort to avoid receiving electrical shocks but did not change behavior meaningfully when effort exertion would confer additional rewards.

The manuscript is well-written and tackles an important question that perfectly fits the scope of the journal and should be of great interest to its readership. I enjoyed reading the manuscript and I do not have any serious issues with the design that would require substantial changes before publication. Nonetheless, I have a number of questions/concerns that I think could use clarification. I will list my comments separately for each section (order does not relate to severity of the comment).

Introduction:

Overall, I think the introduction is very clear. I only have two minor comments.

R3, Comment 1. - The authors, throughout the manuscript, frequently use the term 'energy', often interchangeably with effort or resources. I do understand the (very intuitive) appeal but it is unclear what energy specifically refers to with respect to effort-based choices. Admittedly, this problem is bigger when talking about mental effort but even in the physical domain, it is still debatable what these energetic resources or costs really represent (e.g., ATP in muscle cells? Glucose? The feeling of fatigue?). Thus, I would personally refrain from using 'energy' and replace it either with more specific terms or other general terms that do not necessarily imply a metabolic cost. That being said, this is not a hill I am willing to die on, so if the authors think 'energy' is the best word to use, then they are of course free to do it.

- **Reply:** We have removed almost all references to "energy" in the abstract and main text, except for a few places in the manuscript where we try to make a general connection between energy expenditure and physical effort.

R3, Comment 2. - The authors end the introduction with a summary of their results. This is unusual (although it looks like it has become more common recently) and maybe something the journal encourages (?), but I do not think it's necessary to foreshadow the rest of the paper and could be taken out. But again, this is a decision I will ultimately leave to the authors and the editor.

- **Reply:** We thank Reviewer 3 for pointing this out. After double-checking the Editorial Request Table, we decided to remove the results summary at the end of the introduction.

Methods:

R3, Comment 3. - As far as I could see, there was no specific sample size justification provided by the authors. How did they arrive at $n = 80$ (and $n = 84$) as their target sample size?

- **Reply:** We note that this comment is similar to **R2/Comment 1**. We now clarify that our sample size was based on the statistical power to detect more general between and within-experiment effects of acute stress on choice behaviour. We have added a power calculation to the "Participants" section to demonstrate this. We also added a statement to the "Limitations" section to clarify that – in light of the sample size – higher-order interactions should be interpreted with caution. To limit redundancy, we kindly refer Reviewer 3 to **R2/Comment 1** for the exact text that was added.

R3, Comment 4. - On page 6, the authors use the terms SCORT and sAA without properly introducing these abbreviations before.

- **Reply:** Thank you for noticing – updated.

R3, Comment 5. - In line 148, the authors state that they measured heart rate at -10 and 0 min (in relation to the MAST). Were these the only times heart rate was measured? If so, why was it not measured after stress? (Although the timing might be in relation to stress offset instead of onset? see my comments below).

- **Reply:** We apologize for the confusion: all timepoints (in digits or as “txx”) are relative to stress offset/start of the effort-based decision-making paradigm. T=0 marks stress offset, negative/positive timepoints refer to measurement times before/after stress offset. We have now thoroughly revised the text in all relevant figure legends. We have also made the following changes to the “Acute Stress Induction” section:

“Psychological and physiological stress measurements were collected at multiple timepoints (note: all timepoints mentioned below are relative to the end of MAST/start of the effort-based decision-making paradigm at t=0). In both experiments, saliva samples were obtained at seven timepoints. Two baseline samples were collected prior to starting the MAST control/stress-induction procedures; one at t=-40 (minutes) and one at t=-10, with MAST instructions shared 5 minutes before the t=-10 sample and the MAST procedures starting immediately after participants had provided the t=-10 sample. The remaining five samples were collected in 10-minute intervals, starting immediately post-MAST/at start of the effort-based decision-making paradigm (i.e., t=0 until t=+40). All samples were collected using Salivette® swabs (Sarstedt, Etten-Leur, the Netherlands) during a 3-min. sampling period and stored at -20°C until analysis.”

R3, Comment 6. - The authors used a novel task to test their primary hypothesis. While I appreciate the overall task design, one of my bigger concerns about this study relates to the interpretation of the results from this task as well as how they relate to those of experiment 2. Specifically, in the ‘thunderstorm’ task, participants are asked to choose between exerting effort or receiving an electrical shock with a certain probability. Given that the shock, in contrast to the physical demand, has a constant intensity and only varies in its probability, could it be possible that stress did not (only) change participants’ evaluation of effort but how they process these probabilities? I know the authors discuss this to a degree, namely in terms of a lack of evidence for changes in (monetary) risk taking under stress. However, some studies do suggest that this can be the case (e.g., Starcke et al., 2008; Nowacki et al., 2019; Pighin et al., 2020), so I do not think it can be fully excluded as a potential explanation for the results. I was wondering if the authors had specific reasons to use shock probability instead of intensity as an independent variable here.

R3, Comment 7. - Moreover, the authors suggest that the results obtained in experiment 2 indicate that stress selectively increases motivation to exert effort in the aversive context of expecting shocks but not in a context in which participants exert effort to obtain rewards. However, I am not sure that the results can be compared that easily. In contrast to the shocks in experiment 1, rewards in experiment 2 were varied by amount (or reward intensity) but not by the probability of receiving them. In other words, the effort-associated rewards and punishments were manipulated in distinct ways across both experiments, namely outcome magnitude vs. outcome probability, respectively. Could it be possible that stress might have affected effort-based decisions about reward if the authors had used a fixed amount (e.g., 15 cents) and varied the probability of receiving it? To be clear, I do not necessarily doubt the validity of each experiment’s finding but think that the conclusion that stress only specifically affect motivation to exert effort in one but not the other context might not be fully warranted due to the differences in task design outlined here.

- **Reply (to Comment 6 and 7):** We thank Reviewer 3 for insightful comments regarding paradigm design. We have chosen to address these comments in a single reply. We will start by providing a rationale for using shock probability in Experiment 1, followed by our reading of the research on risk perception (and related domains) under stress, and will end with potentially different effects of acute stress on probability versus magnitude computations.
 - Our decision to vary shock probability (rather than magnitude) in Experiment 1 was based on the notion that threat-of-shock paradigms commonly rely on probabilistic administration of a shock with fixed intensity¹⁻³. In contrast, pain studies commonly rely on deterministic administration of a shock with varying intensity^{4,5}. While we believe that this distinction is arbitrary, we decided to stick with shock administration procedures that are more commonly used in threat research. Moreover, probabilistic administration of a shock with fixed intensity circumvents the issue of having to interpret behaviour on low shock intensity trials (e.g., can we even think of a shock with very low intensity as being painful and/or threatening?). We

realize that this design choice results in a situation where the units for shocks and effort differ, but our experiment, as well as an abundance of previous work⁶⁻⁸, shows that humans can make trade offs between incommensurable goods.

- Nevertheless - and this brings us to Reviewer 3's second point - the consequence of this design choice is indeed that Experiment 1 group differences might be explained by stress-induced changes in risk perception. However, our reading of the literature is that our results, if anything, demonstrate the opposite. That is, the effects of acute stress on risk perception are generally in the opposite direction compared to our own results. We have now included a paragraph in the discussion that compares our results to results on risk perception and other decision-making biases related to loss/punishment.
- Finally, another consequence of our design choice is that it makes Experiment 1 and 2 less comparable. Here, we note that this comment is similar to **R2/Comment 7**. We agree that there might be a possibility that acute stress differentially affects probability versus magnitude computations. We now clearly discuss this possibility in the discussion.

In the discussion, the following text related to risk perception has now been added:

“Another possibility is that our acute stress challenge may have impacted decision-making in the context of risk, uncertainty and/or potential punishments. Previous work has investigated how acute stress affects risk-taking behaviour, loss aversion, and ambiguity aversion⁵⁷⁻⁶². Although this work has revealed generally mixed effects that are dependent on individual characteristics and stressor type^{59,62-64}, a number of studies in this domain have demonstrated that acute stress can increase risk-taking behaviour⁶⁵⁻⁶⁸. Moreover, combined glucocorticoid and noradrenergic activation, as is common during acute stress, can reduce aversion to losses⁶¹. Given that acutely stressed participants were more, and not less, likely to exert effort to neutralize a probability of shock, and in light of the above-mentioned results, stress-induced increases in risk taking and/or reductions in loss aversion cannot fully account for Experiment 1 results.”

In the discussion, the following text related to Experiment 1-2 differences has now been added:

“At the same time, the importance of between-study differences must also be acknowledged, given that the exact subjective value computations employed strongly depend on the type of effort^{18,19}, the other option on offer⁴⁰, outcome valence^{71,72}, and interindividual differences (e.g., in aversion to effort, reward/punishment sensitivity, and fear of aversive outcomes^{40,59}). On this note, it is worth mentioning that Experiment 1 and 2 differed not only in reinforcer type (i.e., shock versus reward), but also in the way that these variables were manipulated. Specifically, Experiment 1 used varying probabilities of experiencing a shock with pre-calibrated intensity, whereas in Experiment 2 guaranteed rewards varied in magnitude. While our findings provide evidence for a stress-induced increase in the willingness to exert effort in the service of avoiding threats, there is the possibility that Experiment 1-2 differences may, at least partially, stem from divergent effects of acute stress on decisions involving threat probability versus reward magnitude. Indeed, some evidence points toward reduced reward sensitivity under stress as measured by reward learning and motivation paradigms⁷³⁻⁷⁵, again underscoring the possibility of reinforcer and/or situationally-specific effects of acute stress on the willingness to exert effort.”

R3, Comment 8. - In addition to the point above, the strength of the conclusion that stress affects aversive but not appetitive motivation suffers a bit from the fact that the two experiments used different participants. It is possible that choice behavior of individuals participating in experiment 1 was simply more sensitive to stress than the choice behavior of participants in experiment 2.

- **Reply:** We appreciate this point, but given 1) the highly similar stress-induction results, 2) the fact that Experiment 1 and 2 participants were sampled from the same population, and, 3) that Experiment 1 and 2 participants, therefore, did not meaningfully differ on self-report measures of chronic stress (PSS: $t_{(162)}=0.00$, 95% CI = -1.01 to 1.00, $p=0.99$), behavioural activation (BAS drive $t_{(162)}=0.17$, 95% CI = -0.14 to 0.17, $p=0.86$) or behavioural inhibition (BIS $t_{(162)}=-0.23$, 95% CI = -0.10 to 0.08, $p=0.82$), we believe that this is an unlikely scenario.

R3, Comment 9. - From the authors' descriptions, I did not fully understand at which timepoint, relative to the MAST, the 'thunderstorm' task was administered. Given the temporal profile of the physiological stress response(s), this information is crucial to discern how exactly stress may have affected behavior. It might be helpful to highlight (maybe in the lower panel in figure 2) when exactly the task took place. Similarly, it should be clarified when the MAST started and ended. I initially assumed 't+00' would indicate the time of stress onset but given the apparent difference in cortisol at that time and that heart rate measures are labeled as pre/post in the figure but as 't-10' and 't+00' in the text, it is possible that 't+00' actually refers to stress offset. Is that correct?

- **Reply:** The Reviewer is correct, and we have now made edits to the "Acute Stress Induction" paragraph and figure legends to clarify this. To limit redundancy, we refer Reviewer 3 to **Reviewer 3/Comment 5**, where the exact changes are listed.

R3, Comment 10. - To further elaborate this point, the authors state that the task is supposed to take place in the 'fight-or-flight' stage of the stress response but do not specify what they mean by that. Usually, this term is primarily associated with the initial SAM-dependent response resulting in increased secretion of catecholamines but here it might be meant to extend to encompass HPA axis activation as well? The distinction is important because SAM activation, as the term 'fight-or-flight' indicates, is thought to facilitate the immediate response to a stressor and normalizes almost instantly after the organism has escaped (or defeated) the threat (which may explain the lack of sAA differences pre/post MAST here). In contrast, glucocorticoid actions are more thought to handle the aftermath of stressful experiences. Although there's an overlap between both systems during the down ramping of the SAM and the up ramping of the HPA axis, this time-window is very brief. The task, on the other hand, took 27 minutes in total and was presumably completed outside the specific context of the MAST (i.e., after a short delay to collect physiological measures and maybe even in a different room?). How did the authors make sure the effects can mostly be related to the early 'fight-or-flight' stage of the stress response? Either way, I think those timing issues and their potential consequences for the mechanisms underlying the observed stress effect could be discussed in a bit more detail.

- **Reply:** Reviewer 3 correctly mentions that we cannot guarantee that all data were collected in the immediate aftermath of the acute stress induction procedure – even despite our attempts to optimize the design to measure immediately post-MAST (e.g., moving directly to an adjacent behavioral lab post-MAST, reducing task duration). In the Limitations section, we now discuss our study in the context of the two stages of an acute stress response:

"While the current study yields novel insights into the motivational processes impacted by acute stress, it is important to acknowledge outstanding questions and limitations that will need to be addressed in future research. First, our study design did not allow us to precisely differentiate between stress phases. The initial "stress reactivity" phase coincides with the almost-immediate and brief release of catecholamines and gradual rise of cortisol, while "stress recovery" is thought to occur during the gradual decrease of cortisol levels back to pre-stressor baseline^{11,76}. Because all participants started the effort-based decision-making task immediately post-MAST, the great majority of task trials were completed at an early stage of the stress response, during which we also observed heightened (adrenaline-mediated) arousal and rising cortisol levels. However, a study design that allows for a more precise distinction between stress phases, or one that can disentangle the role of catecholamines versus glucocorticoids, will provide a deeper understanding of stress phase-dependent changes in the willingness to exert effort under threat."

Results:

R3, Comment 11. - Line 396: the authors refer to figure 1A but I think this should be 3A.

- **Reply:** We apologize for the confusion - we referenced Figure 1A to indicate that we looked at 4 separate quadrants in the sampling space. We have now edited the text to make this clearer. Text now reads:

"see Figure 1A for quadrants"

R3, Comment 12. - Why did the authors choose to group the different effort/threat-of-shock combinations into a 2x2 comparison matrix for their post-hoc analyses instead of looking at all 4x4 comparisons (figure 3B)? Was it to increase the amount of data that goes into each comparison or maybe to reduce the amount of post-hoc tests to correct for in the analysis? I do not have a problem with any of these approaches, but it should be stated for transparency and reproducibility purposes.

- **Reply:** This was exactly our intention. First, we do not believe that condition differences are interesting if they cannot be observed across a reasonable part of the sampling space. Secondly, given the relatively low number of trial repetitions, and the large amount of offers, we are not adequately powered to investigate condition differences at the level of individual offers anyway. We now clarify this rationale in the “Statistical Analyses” section:

“Observed (Condition×Effort cost×Threat-of-shock) interactions in Experiment 1 were followed up by a post hoc test assessing the presence of an Effort Cost×Threat-of-shock interaction for each condition separately. To further break down this interaction, we assessed condition differences for four distinct offer types (i.e., high/low effort/threat; Figure 1A). This approach allowed us to uncover condition differences that were observable across a reasonable part of the offer sampling space (~25%), thereby limiting the amount of tests, and thus the false positive rate.”

R3, Comment 13. - Looking at Figure 3B, the largest differences (and probably the driving factor of the significant result) are found in the 80% effort/20% threat-of-shock bin and the 80% effort/30% threat-of-shock bin, where the stress group displayed much higher motivation to exert the effort. More generally, the differences between the groups seems to be most pronounced at intermediate effort levels (60 and 80%) instead of the high effort/low probability end of the scale as discussed by the authors (e.g., the 100% effort/20 and 100% effort/30% probability bins are almost identical between groups). To me, the stronger sensitivity to stress particularly in these in-between demand levels could suggest that participants operated close to their individual indifference points in these trials. As such, it could be argued that stress specifically tips the scale in situations in which participants have a hard time deciding between options. I would love to read the authors’ thoughts on this slightly different interpretation of their findings.

- **Reply:** Intuitive as it is, we believe that our results do not support this interpretation. To illustrate: the top-3 largest acceptance difference between $MAST_{PLC}$ and $MAST_{EXP}$ groups was found for 80% effort/20% threat-of-shock (average difference of 0.20), 80% effort/30% threat of shock (average difference of 0.16) and 60% effort/20% threat-of-shock (average difference of 0.13). However, acceptance rates for each group were mostly far away from objective (i.e., 50%) indifference points for 80% effort/20% threat-of-shock ($MAST_{PLC}=0.12$, $MAST_{EXP}=0.32$), 80% effort/30% threat-of-shock ($MAST_{PLC}=0.37$, $MAST_{EXP}=0.53$), or for 60% effort/20% threat-of-shock ($MAST_{PLC}=0.27$, $MAST_{EXP}=0.40$). Thus, although threat and effort levels are quite closely matched for these offers, the empirical data do not suggest that participants had a hard time deciding about these offers. In fact, if anything, $MAST_{EXP}$ participants moved into the direction of indifference for these offers.

R3, Comment 14. - For experiment 2, the authors report main effects and (non-significant) condition-related interaction effects, but no other tested effects. I understand that the main text would become difficult to read if effects from all predictors were included but it would be great to add a (supplemental?) table that lists all main and interaction effects from the regression models used in the study (including experiment 1).

- **Reply:** A new table (Supplemental Table 3) has been added. This table contains all output from key regression models used in Experiment 1, Experiment 2, and Experiment 1-vs-2 analyses.

R3, Comment 15 - With respect to the statistical comparison of the stress effects between studies, the authors state that stress only increased motivation in the threat but not the reward context. At the same time, they acknowledge that under reward prospect, individuals were more likely to exert effort in general. Is it possible that the lack of increase in effort exertion under stress in experiment 2 simply reflects ceiling effects?

R3, Comment 16. - Although not significant, it struck me as interesting that descriptively, stress seemed to reduce motivation to exert effort in experiment 2, which would fit other work showing increased effort aversion under stress. In line with experiment 1, the strongest negative effect was again at a more intermediate effort level (80% effort/5 cents reward). I would love to read the authors' thoughts on this (again, being aware that this does not come out as statistically significant).

- **Reply (to Comment 15 and 16):** We have considered the possibility that experiment design features might account for the absence of a stress effect in Experiment 2, but believe that this is an unlikely scenario. First, we carried out various sensitivity analyses (e.g., leaving out Experiment 2 inflexible responders, leaving out offer types with generally high acceptance rates) which did not change the results. Secondly, if anything, Experiment 2 suggests that MAST_{EXP} participants were overall less likely to exert effort. These results indeed hint at the possibility that acute stress might reduce the motivation to exert effort in exchange for reward, but we feel that Experiment 2 results are not convincing enough to make this argument. In addition, the Editor recommended that we emphasize Experiment 1 results in the discussion. As a compromise, we have decided to more generally touch upon this possibility (i.e., in the context of Experiment 1-2 differences), rather than using the weak numerical trend in Experiment 2 specifically:

“While our findings provide evidence for a stress-induced increase in the willingness to exert effort in the service of avoiding threats, there is the possibility that Experiment 1-2 differences may, at least partially, stem from divergent effects of acute stress on decisions involving threat probability versus reward magnitude. Indeed, some evidence points toward reduced reward sensitivity under stress as measured by reward learning and motivation paradigms⁷³⁻⁷⁵, again underscoring the possibility of reinforcer and/or situationally-specific effects of acute stress on the willingness to exert effort.”

Discussion:

R3, Comment 17. - Lines 525/526: The authors state that ‘Moreover, following the decision to exert effort, the same participants exerted greater force intensity while trying to neutralize the threat-of-shock.’ From the results, this looked like a simple group difference comparison. Was it indeed the very same participants (i.e., was there a correlational relationship where participants with stronger effort aversion were the ones displaying the highest force intensity)?

- **Reply:** This phrasing is indeed confusing, and we have now updated this sentence (to “**these participants**”). Reviewer 3 is correct in assuming that we only investigated the main effect of Condition in force data analyses, given that the missingness in these data are not random. That is, the availability of force data is fully dependent on the offer because participants only squeeze for offers that they feel are “worth it”. This complicates investigation of choice-force relationships (e.g., between general offer acceptance tendencies and force intensity), because people with stronger effort aversion will – by definition – have less force data. Notwithstanding this limitation, we have calculated the correlation between average effort acceptance and average force intensity (which shows a numerical trend in the other direction: $r_{(78)}=0.10$, 95% CI = -0.12 to 0.32, $p=0.36$), with the important caveat that such analyses do not control for the type of offers that were accepted. In light of the above considerations, we believe that Experiment 1-2 data are not well-suited to investigate choice-force relationships, but if the Reviewer believes this result is valuable, we are willing to include it in the manuscript.

R3, Comment 18. - Line 529: ‘promotes’ should be ‘promote’

- **Reply:** Thank you – adjusted.

References (Rebuttal)

1. Binti Affandi AH, Pike AC, Robinson OJ. (2022). Threat of shock promotes passive avoidance, but not active avoidance. *Eur J Neurosci*, 55(9): 2571–2580. <https://doi.org/10.1111/ejn.15184>
2. Hulsman, A. M., Kaldewaij, R., Hashemi, M. M., Zhang, W., Koch, S. B. J., Figner, B., Roelofs, K., & Klumpers, F. (2021). Individual differences in costly fearful avoidance and the relation to psychophysiology. *Behaviour Research and Therapy*, 137, 103788. <https://doi.org/10.1016/j.brat.2020.103788>
3. Sambrano, D.C., Lormestoire, A., Raio, C. *et al.* (2022). Neither Threat of Shock nor Acute Psychosocial Stress Affects Ambiguity Attitudes. *Affec Sci*, 3, 425–437 <https://doi.org/10.1007/s42761-022-00109-6>
4. Vogel, T. A., Savelson, Z. M., Otto, A. R., & Roy, M. (2020). Forced choices reveal a trade-off between cognitive effort and physical pain. *eLife*, 9. <https://doi.org/10.7554/elife.59410>
5. Coll, M. P., Slimani, H., Woo, C. W., Wager, T. D., Rainville, P., Vachon-Preseau, É., & Roy, M. (2022). The neural signature of the decision value of future pain. *Proceedings of the National Academy of Sciences of the United States of America*, 119(23), e2119931119. <https://doi.org/10.1073/pnas.2119931119>
6. Chong, T. T., Apps, M. a. J., Giehl, K., Sillence, A., Grima, L. L., & Husain, M. (2017). Neurocomputational mechanisms underlying subjective valuation of effort costs. *PLOS Biology*, 15(2), e1002598. <https://doi.org/10.1371/journal.pbio.1002598>
7. Prévost, C., Pessiglione, M., Météreau, E., Cléry-Melin, M., & Dreher, J. (2010). Separate valuation subsystems for delay and effort decision costs. *The Journal of Neuroscience*, 30(42), 14080–14090. <https://doi.org/10.1523/jneurosci.2752-10.2010>
8. Levy, D. J., & Glimcher, P. W. (2012). The root of all value: a neural common currency for choice. *Current Opinion in Neurobiology*, 22(6), 1027–1038. <https://doi.org/10.1016/j.conb.2012.06.001>

2nd May 24

Dear Dr Hernaus,

Your manuscript titled "The Drive to Survive: Acute Stress Promotes Aversive Motivation" has now been seen by our reviewers, whose comments appear below. In light of their advice I am delighted to say that we are happy, in principle, to publish a suitably revised version in *Communications Psychology* under the open access CC BY license (Creative Commons Attribution v4.0 International License).

We therefore invite you to revise your paper one last time to address an editorial concern and a list of editorial requests. At the same time we ask that you edit your manuscript to comply with our format requirements and to maximise the accessibility and therefore the impact of your work.

EDITORIAL REQUESTS:

Editorially, we ask that you revise the presented power analysis As reviewer#3 points out the power analysis that allows to determine the sample size, should have been run a priori before the data collection. Post-hoc power analyses are strongly discouraged by the journal. In the absence of a prior power analysis, the most suitable solution is to run a "power sensitivity analysis" which consists of determining what is the smallest observable effect size with enough power given your sample size and your design (see Lakens, Collabra 2022). Editorially, we ask that you replace the post-hoc power analysis with a sensitivity analysis unless the power analysis was indeed run a priori.

SUBMISSION INFORMATION:

OPEN ACCESS:

Communications Psychology is a fully open access journal. Articles are made freely accessible on publication under a CC BY license (Creative Commons Attribution 4.0 International License). This license allows maximum dissemination and re-use of open access materials and is preferred by many research funding bodies.

For further information about article processing charges, open access funding, and advice and support from Nature Research, please visit <https://www.nature.com/commspsychol/article-processing-charges>

At acceptance, you will be provided with instructions for completing this CC BY license on behalf of all authors. This grants us the necessary permissions to publish your paper. Additionally, you will be asked to declare that all required third party permissions have been obtained, and to provide billing information in order to pay the article-processing charge (APC).

* **DATA AVAILABILITY:**

[link redacted]

Best regards,

Jennifer Bellingtier

Jennifer Bellingtier, PhD

Senior Editor

Communications Psychology

Eva R. Pool, PhD

Editorial Board Member

Communications Psychology

orcid.org/0000-0001-5929-1007

REVIEWER EXPERTISE:

Reviewer #1 Stress and Cognition

Reviewer #2 Stress, Motivation and decision making

Reviewer #3 Computational modeling of decision making

REVIEWERS' COMMENTS:

Reviewer #1 (Remarks to the Author):

Thank you very much for the resubmission. All my comments have been addressed. The adjustments in response to the editorial and other reviewer comments are very clear and improved the manuscript. I have no further comments and endorse publication of this manuscript.

Reviewer #2 (Remarks to the Author):

I want to thank the reviewers for this thoroughly revised version of the manuscript. My main concerns were addressed adequately and in sufficient detail. The addition of the limitations and power analysis strengthens the data and helps to contextualize the findings. I have no further hesitation in publishing this data. Lastly, I want to point out that this manuscript made for a very enjoyable and informative read. I believe this research adds to a growing literature on the relationship between motivation, effort, and stress—as such, it makes important contributions that will resonate with the field and result in exciting follow-up research.

Reviewer #3 (Remarks to the Author):

I thank the authors for their comprehensive and thoughtful replies and edits regarding my concerns about their initial manuscript draft. I think they have done a phenomenal job and I am fully content with the changes they made!

The only (very small) thing I would add: The authors have included a thorough sample-size justification. I am not entirely sure if this was performed prior to study start (and simply not fully reported in the initial

manuscript) or whether these calculations were made only in response to the reviewers' comments. If it's the latter, I would specify that these are not a priori power analyses.

Authors reply to Reviewer Comments on COMMSPSYCHOL-23-0396 (revision 2).

We thank the Editor and Reviewers for their kind comments. Below, we respond to the final point raised by R3.

REVIEWERS' COMMENTS:

Reviewer #1

Thank you very much for the resubmission. All my comments have been addressed. The adjustments in response to the editorial and other reviewer comments are very clear and improved the manuscript. I have no further comments and endorse publication of this manuscript.

Reviewer #2

I want to thank the reviewers for this thoroughly revised version of the manuscript. My main concerns were addressed adequately and in sufficient detail. The addition of the limitations and power analysis strengthens the data and helps to contextualize the findings. I have no further hesitation in publishing this data. Lastly, I want to point out that this manuscript made for a very enjoyable and informative read. I believe this research adds to a growing literature on the relationship between motivation, effort, and stress—as such, it makes important contributions that will resonate with the field and result in exciting follow-up research.

Reviewer #3

I thank the authors for their comprehensive and thoughtful replies and edits regarding my concerns about their initial manuscript draft. I think they have done a phenomenal job and I am fully content with the changes they made!

R3, Comment 1:

The only (very small) thing I would add: The authors have included a thorough sample-size justification. I am not entirely sure if this was performed prior to study start (and simply not fully reported in the initial manuscript) or whether these calculations were made only in response to the reviewers' comments. If it's the latter, I would specify that these are not a priori power analyses.

- **Reply:** The power calculations were indeed *a priori*, based on our previous work. This was already mentioned in previous versions of the Reporting Summary, but in the new version of this document – and in the main text -we have made this point even clearer.

In Reporting Summary:

“*A priori* power calculations aimed to ensure sufficient power to detect more general medium effects of acute stress on the willingness to exert effort within and between experiments, with reduced power to detect more complex higher-order interactions with a similar effect size (e.g., with total $n=160$: power=0.88 to detect an Experiment \times Condition interaction on choice behaviour with effect size $\eta^2=0.065$ at $\alpha=0.05$; power=0.59 to detect a Condition \times Effort \times Threat interaction on choice behaviour). Specifically, the power analysis was based on ensuring power=0.80 to detect a ~10% difference in overall acceptance of effort between acute stress and control participants at $\alpha=0.05$ (two-tailed) and assuming an SD of 12 in each group (group difference and SD based on previous empirical data investigating the effect of acute stress on an effort-based learning task; <https://doi.org/10.1016/j.psyneuen.2021.105646>).”

In Methods/Participants:

“*A priori* sample size determinations were based on the ability to ensure sufficient power to detect more general medium effects of acute stress on the willingness to exert effort within and between experiments, although the power to detect more complex higher-order interactions with a similar effect size was reduced.”